# PART-AWARE PERSONALIZED SEGMENT ANYTHING MODEL FOR PATIENT-SPECIFIC SEGMENTATION

## ABSTRACT

Precision medicine, such as patient-adaptive treatments utilizing medical images, poses new challenges for image segmentation algorithms due to (1) the large variability across different patients and (2) the limited availability of annotated data for each patient. In this work, we propose a data-efficient segmentation method to address these challenges, namely ***Part-aware Personalized Segment Anything Model*** ($\mathbf{P^2SAM}$). Without any model fine-tuning, $P^2SAM$ enables seamless adaptation to any new patients relying only on one-shot patient-specific data. We introduce a novel part-aware prompt mechanism to select multiple-point prompts based on part-level features of the one-shot data, which can be extensively integrated into different promptable segmentation models, such as SAM and SAM 2. To further promote the robustness of the part-aware prompt mechanism, we propose a distribution-similarity-based retrieval approach to determine the optimal number of part-level features for a specific case. $P^2SAM$ improves the performance by +8.0% and +2.0% mean Dice score within two patient-specific segmentation tasks, and exhibits impressive generality across different domains, *e.g.*, +6.4% mIoU on the PerSeg benchmark. Code will be released upon acceptance.

## 1 INTRODUCTION

Advances in modern precision medicine and healthcare have emphasized the importance of personalized treatment, aiming at adapting to the specific patient (Hodson, 2016). For instance, in radiation therapy, patients undergoing multi-fraction treatment would benefit from longitudinal medical data analysis that helps timely adjust treatment planning specific to the individual patient (Sonke et al., 2019). To facilitate the treatment procedure, such analysis demands timely and accurate automatic segmentation of tumors and critical organs from medical images, which has underscored the role of computer vision approaches for medical image segmentation tasks (Hugo et al., 2016; Jha et al., 2020). Despite the great progress made by previous works (Ronneberger et al., 2015; Isensee et al., 2021; Dumitru et al., 2023), their focus remains on improving the segmentation accuracy within a standard paradigm: trained on a large number of annotated data and evaluated on the *in-distribution* validation set. However, personalized treatment presents unique challenges for segmentation algorithms: (1) the large variability across different patients, and (2) the limited availability of annotated training data for each patient. Overcoming these obstacles requires a segmentation approach that can reliably generalize to *out-of-distribution* patients, in a data-efficient manner.

In this work, we address the unmet needs of the patient-specific segmentation by formulating it as an in-context segmentation problem, leveraging the promptable segmentation mechanism inherent in Segment Anything Model (SAM) (Kirillov et al., 2023). Under this objective, our method seamlessly adapts to any new (*out-of-distribution*) patients relying only on one-shot patient-specific prior data without requiring additional training, thus in a data-efficient manner. Moreover, such data can be obtained in a standard clinical protocol (Chen et al., 2023), which will not burden clinical researchers. To this end, we propose $\mathbf{P^2SAM}$: *Part-aware Personalized Segment Anything Model*.

In the original prompt mechanism of SAM, as illustrated by Figure 1, a single-point prompt may result in ambiguous prediction, indicating the limitation in both in-domain and out-of-domain applications (Zhang et al., 2023; Huang et al., 2024). To alleviate the ambiguity problem, following the statement in SAM, "*ambiguity is much rarer with multiple prompts*", we propose a novel part-aware prompt mechanism to meticulously select multiple-point prompts based on part-level features of

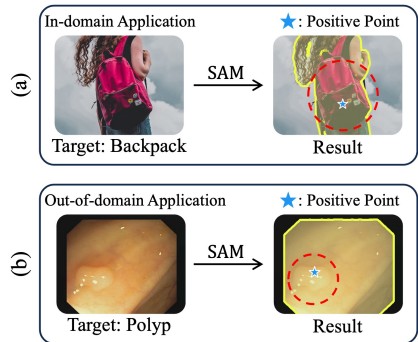 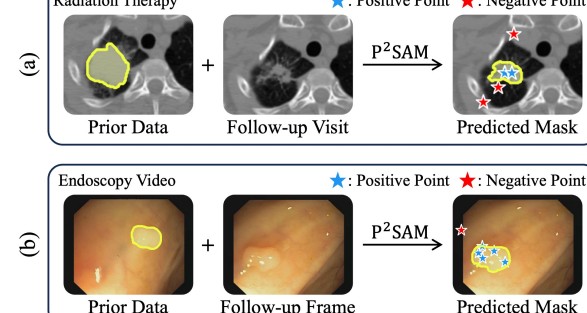

Figure 1: Illustration of SAM's ambiguity problem. The ground truth is circled by a red dashed circle; the predicted mask is depicted by a yellow solid line.

Figure 2: Illustration of two patient-specific segmentation tasks. $P^2$SAM can segment the follow-up data by utilizing the prior data as multiple-point prompts. Prior and predicted masks are depicted by a solid yellow line.

the one-shot prior data. As illustrated in Figure 2, our method enables reliable adaptation to a new patient across various tasks with one-shot prior data. To extract part-level features, we commence by clustering the prior data into multiple groups in the embedding space. Then, we select multiple-point prompts based on the similarity between these part-level features and the follow-up data. The proposed approach can be generalized to different promptable segmentation models that support the point modality, such as SAM and its successor, SAM 2 (Ravi et al., 2024). Here, we primarily utilize SAM as the backbone model, but SAM 2 will be integrated within the specific setting.

On the other hand, when the number of parts is sub-optimal, either more or less, the chance of encountering outlier prompts may increase. An extreme case is to cluster each image patch into different groups, which renders a lot of outlier prompts (Liu et al., 2023). To make the part-aware prompt mechanism more robust, we introduced a retrieval approach to investigate the optimal number of parts required for each case. The retrieval approach is based on the distribution similarity between the foreground feature of the prior data and the result obtained under the current part count. This principle is motivated by the fact that tumors and normal organs manifest in distinct distributions within medical imaging technologies (García-Figueiras et al., 2019).

With the aforementioned designs, $P^2$SAM addresses a general challenge—ambiguity—in promptable segmentation models through a simple yet effective approach, benefiting both medical and natural image domains. The key contributions of this work lie in three-fold:

1. We formulate the patient-specific segmentation as an in-context segmentation problem, resulting in a data-efficient segmentation method, $P^2$SAM, which operates with only one-shot prior data and requires no model fine-tuning.

2. We propose a novel part-aware prompt mechanism that meticulously selects multiple-point prompts based on part-level features, combined with a distribution-similarity-based retrieval approach to determine the optimal number of part-level features for each case. These two designs effectively mitigate the ambiguity problem in promptable segmentation models and enable $P^2$SAM to adapt across different tasks, models, and domains.

3. Our method largely benefits real-world applications like patient-specific segmentation, one-shot segmentation, and personalized segmentation. Experiment results demonstrate that $P^2$SAM improves the performance by $+8.0\%$ and $+2.0\%$ mean Dice score in two patient-specific segmentation tasks and achieves a new state-of-the-art result, *i.e.*, $95.7\%$ mIoU on the personalized segmentation benchmark PerSeg.

## 2 RELATED WORK

**Segmentation Generalist.** Over the past decade, various segmentation tasks including semantic segmentation (Strudel et al., 2021; Li et al., 2023a), instance segmentation (He et al., 2017; Li et al., 2022a; 2023b), and panoptic segmentation (Carion et al., 2020; Cheng et al., 2021b;a; Li et al., 2022b) have been extensively explored for the image and video modalities. Motivated by the suc-

cess of foundational language models (Radford et al., 2018; 2019; Brown et al., 2020; Touvron et al., 2023), the computer vision research community is increasingly paying attention to developing more generalized models that can tackle various vision or multi-modal tasks, or called foundation models (Li et al., 2022b; Oquab et al., 2023; Yan et al., 2023; Wang et al., 2023a;b; Kirillov et al., 2023). Notably, Segment Anything model (SAM) (Kirillov et al., 2023) and its successor, SAM 2 (Ravi et al., 2024) introduces a promptable model architecture, including the positive- and negative-point prompt; the box prompt; and the mask prompt. SAM and SAM 2 emerge with an impressive zero-shot interactive segmentation capability after pre-training on the large-scale dataset. Given the remarkable generalization capacity, researchers within the medical image domain have been seeking to build foundational models for medical image segmentation (Wu et al., 2023; Wong et al., 2023; Wu & Xu, 2024; Zhang & Shen, 2024) upon them. Certain approaches (Ma et al., 2024a;b) have already shown promising results: MedSAM (Ma et al., 2024a) has exhibited significant performance across various medical image segmentation tasks after fine-tuning SAM on an extensive medical dataset. MedSAM 2 (Ma et al., 2024b) incorporates SAM 2 to segment a 3D medical image volume as video. However, whether these methods can achieve zero-shot performance as impressive as SAM and SAM 2 remains an open question that requires further investigation (Ma et al., 2024b).

**In-Context Learning** First introduced as a new paradigm in natural language processing (Brown et al., 2020), in-context learning allows the model to adapt to unseen input patterns with a few prompts and examples, without the need to fine-tune the model. Similar ideas (Li et al., 2023b; Sonke et al., 2019; Rakelly et al., 2018) have been explored in other fields. In computer vision, few-shot segmentation (Rakelly et al., 2018; Wang et al., 2019b; Liu et al., 2020; Leng et al., 2024), like PANet (Wang et al., 2019b), aims to segment new classes with only a few examples; in adaptive therapy (Sonke et al., 2019), several works (Wang et al., 2019a; Elmahdy et al., 2020; Wang et al., 2020; Chen et al., 2023) attempt to leverage limited patient-specific data to adapt a model to new patients, but these methods still require model fine-tuning in different manners. Recent advancements, such as Painter (Wang et al., 2023a) and SegGPT (Wang et al., 2023b) pioneer novel in-context learning approaches for vision tasks, enabling the timely segmentation of images based on specified image-mask prompts. SEEM (Zou et al., 2024) further explores this concept by investigating different prompt modalities. More recently, PerSAM (Zhang et al., 2023) and Matcher (Liu et al., 2023) have utilized SAM to tackle few-shot segmentation through the in-context learning fashion. PerSAM introduces a novel task, known as personalized object segmentation (Zhang et al., 2023), which aims at adapting SAM to new views of a specific object. However, PerSAM prompts SAM with only a singular prompt, leading to the ambiguity problem (Kirillov et al., 2023) in the segmentation results. On the other hand, Matcher enhances segmentation accuracy by utilizing multiple sets of point prompts. However, Matcher's prompt generation mechanism is based on patch-level features. This mechanism makes Matcher dependent on DINOv2 (Oquab et al., 2023) to generate prompts, which is particularly pre-trained under a patch-level objective. Despite this, Matcher still generates a lot of outlier prompts. Thus, Matcher relies on a complicated framework and lacks flexibility and robustness when integrated into other vision backbones, including SAM.

## 3 METHOD

We first introduce the problem setting within the context of patient-specific segmentation in Section 3.1. We introduce our proposed methodology, $P^2SAM$, in Section 3.2. Note that our method can adapt to various domains. Therefore, we incorporate natural image illustrations in this section to provide a more intuitive understanding. Finally, we present an optional fine-tuning strategy in Section 3.3, to adapt the backbone model to the medical image domain if required.

### 3.1 PROBLEM SETTING

Our method aims to adapt a promptable segmentation model to *out-of-distribution* patients, with only one-shot patient-specific prior data. As shown in Figure 2, such data can be obtained in a standard clinical protocol, either from the initial visit of radiation therapy or the first frame of medical video. The prior data includes a reference image $I_R$ and a mask $M_R$ delineating the segmented object. Given a target image, $I_T$, our goal is to predict its mask $M_T$, without additional human annotation costs or model training burdens. This setting is also suitable for object-specific segmentation, where the target image represents a new view of the same object depicted in the prior data.

Figure 3: Illustration of presenting the prior data as multiple-point prompts. Masks are depicted by a yellow solid line. We first cluster foreground features in the reference image into part-level features. Then, we select multiple-point prompts based on the cosine similarity ($\otimes$ in the figure) between these part-level features and target image features. A colorful star, matching the color of the corresponding part, denotes a positive-point prompt, while a gray star denotes a negative-point prompt. These prompts are subsequently fed into the promptable decoder to do prediction.

## 3.2 METHODOLOGY OVERVIEW

**Part-aware Prompt Mechanism.** To facilitate a clearer understanding of the significance of each part in our part-aware prompt mechanism, we illustrate this approach using a natural image, as shown in Figure 3. We utilize SAM (Kirillov et al., 2023) as the backbone model here, but our approach can be generalized to other promptable segmentation models, such as SAM 2 (Ravi et al., 2024), as long as they support the point prompt modality. Given the reference image-mask pair from the prior data, $\{I_R, M_R\}$, P²SAM first apply SAM's *Encoder* to extract the visual features $F_R \in \mathbb{R}^{h \times w \times d}$ from the reference image $I_R$. Then, we utilize the reference mask $M_R$ to select foreground features $F_R^f$ according to:

$$F_R^f = F_R \circ M_R \tag{1}$$

where $\circ$ represents the mask selection, $F_R^f \in \mathbb{R}^{n_f \times d}$, and $n_f$ represents the number of foreground features. We further cluster $F_R^f$ with $k$-mean++ (Arthur et al., 2007) into $n$ parts. Here, we showcase an example of $n=4$. We obtain the centroid of each part as the representative for the part-level features, by applying an average pooling, denoting as $\{P_R^c\}_{c=1}^n \in \mathbb{R}^{n \times d}$. For illustration, we align the features of each part with pixels in the RGB space, thereby contouring the corresponding regions for each part in the image, respectively. We observe that SAM's encoder tends to cluster features together based on texture features, such as the characters and images depicted on the can.

After that, we extract the features $F_T$ from the target image $I_T$ using the same *Encoder*, and compute similarity maps $\{S^c\}_{c=1}^n \in \mathbb{R}^{n \times h \times w}$ based on the cosine similarity between the extracted part-level features $\{P_R^c\}_{c=1}^n$ and $F_T$ by:

$$S^c{}_{ij} = \frac{P_R^c \cdot F_{Tij}}{\|P_R^c\|_2 \cdot \|F_{Tij}\|_2} \tag{2}$$

We determine $n$ positive-point prompts $\{Pos^c\}_{c=1}^n$ with the highest similarity score on each similarity map $S^c$, depicted as colorful stars in Figure 3.

For natural images, the background of the reference image and the target image may exhibit little correlation. Thus, following the approach in PerSAM (Zhang et al., 2023), we choose one negative-point prompt $\{Neg\}$ with the lowest score on the average similarity map $\frac{1}{n}\sum_{c=1}^n S^c$. $\{Neg\}$ is depicted as the gray star in Figure 3. However, for medical images, the background of the reference image is highly correlated with the background of the target image, usually both representing normal anatomical structures. As a result, in medical images, shown as Figure 2 in Section 1, we identify multiple negative-point prompts $\{Neg^c\}_{c=1}^n$ from the background. This procedure mirrors the selection of multiple positive-point prompts but we use background features $F_R^b$ by replacing $M_R$ with its logical negation $\widetilde{M_R}$ in Equation 1. Finally, we send both positive- and negative-point prompts into SAM's *Promptable Decoder* and get the predicted mask $M_T$ for the target image.

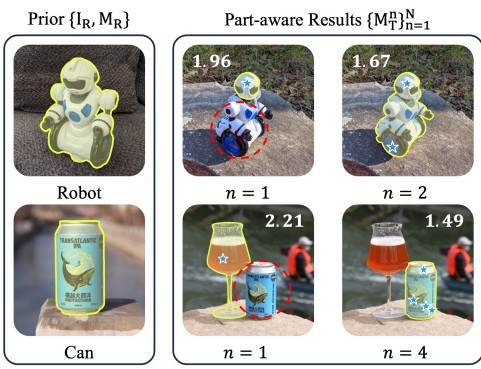 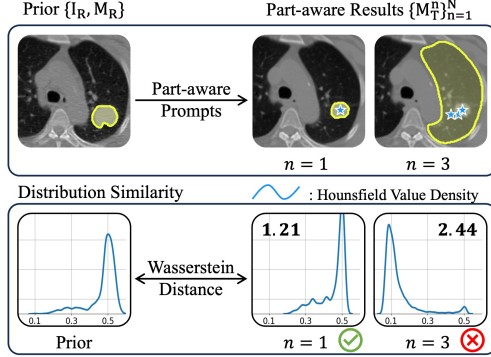

Figure 4: Illustration of P$^2$SAM's improvement. Blue stars represent positive-point prompts.

Figure 5: Illustration of the approach to retrieve the optimal number of parts for a specific case.

**Retrieve the Optimal Number of Parts.** Improvements of the part-aware prompt mechanism are illustrated in Figure 4. The proposed approach can naturally avoid the ambiguous prediction introduced by SAM (*e.g.*, robot) and also improve precision (*e.g.*, can). However, this approach may occasionally result in outliers, as observed in the segmentation example in Figure 5, $n$=3. Therefore, we propose a distribution-similarity-based retrieval approach to answer the question, "*How many part-level features should we choose for each case?*". We assume the correct target foreground feature $F_T^f = F_T \circ M_T$, and the reference foreground feature $F_R^f$ should belong to the same distribution. This assumption is grounded in the fact that tumors and normal organs will be reflected in distinct distributions by medical imaging technologies (García-Figueiras et al., 2019), also observed by the density of Hounsfield Unit value in Figure 5. To retrieve the optimal number of parts for a specific case, we first define $N$ different part counts, $n \in \{1, \cdots, N\}$, and obtain $N$ sets of part-aware target foreground features $\{\{F_T^f\}^n\}_{n=1}^N$. Following WGAN (Arjovsky et al., 2017), we utilize Wasserstein distance to measure the distribution similarity between the reference foreground feature $F_R^f$ and each target foreground feature $\{F_T^f\}^n$. We determine the final number of part-level features, $n$, with the smallest distance. The smaller distance value for the correct prediction in Figure 4 indicates this approach can be further extended to natural images.

### 3.3 Adapt SAM to Medical Image Domain if Needed

Segment Anything Model (SAM) (Kirillov et al., 2023) is initially pre-trained on the SA-1B dataset. Despite the large scale, a notable domain gap persists between natural and medical images. In more realistic medical scenarios, clinic researchers could have access to certain public datasets (Aerts et al., 2015; Jha et al., 2020) tailored to specific applications, enabling them to fine-tune the model. Nevertheless, even after fine-tuning, the model can still be limited to generalize across various *out-of-distribution* medical data from different institutions because of the large variability in patient population, demographics, imaging protocol, etc., as mentioned in Section 1. P$^2$SAM can then be flexibly plugged into the fine-tuned model to enhance robustness on testing cases.

Specifically, when demanded, we utilize *in-distribution* datasets (Aerts et al., 2015; Jha et al., 2020) to adapt SAM into the medical image domain. We try full fine-tune, and Low-Rank adaptation (LoRA) (Hu et al., 2021) for further efficiency. During the fine-tuning, similar to Med-SA (Wu et al., 2023), we adhere closely to the interactive training strategy outlined in SAM to maintain the interactive ability. Details can be found in Appendix B. Then, we employ *out-of-distribution* datasets (Bernal et al., 2015; Hugo et al., 2016) obtained from various institutions to mimic new patient cases. Note that there is no further fine-tuning on these datasets.

## 4 Experiments

We first introduce our experimental settings in Section 4.1. Then we evaluate the quantitative results of our approach in Section 4.2. We show qualitative results in Section 4.3. Finally, we conducted several ablation studies to investigate our designs in Section 4.4.

## 4.1 EXPERIMENT SETTINGS

**Dataset.** We utilize a total of four medical datasets, including two *in-distribution (i.d.)* datasets: The NSCLC-Radiomics dataset (Aerts et al., 2015), collected for non-small cell lung cancer (NSCLC) segmentation, contains data from 422 patients. Each patient has a 3-dimensional computed tomography volume along with corresponding segmentation annotations. The Kvasir-SEG dataset (Jha et al., 2020), contains 1000 labeled endoscopy polyp images, with different resolutions ranging from $332 \times 487$ to $1920 \times 1072$. Two *out-of-distribution (o.o.d)* datasets from different institutions: The 4D-Lung dataset (Hugo et al., 2016), collected for longitudinal analysis, contains data from 20 patients, within which 13 patients underwent multiple visits, 3 to 8 visits for each patient. For each visit, a 3-dimensional computed tomography volume along with corresponding segmentation labels is available. The CVC-ClinicDB dataset (Bernal et al., 2015), contains 612 labeled polyp images selected from 29 endoscopy videos, with a resolution of $384 \times 288$. *i.d.* datasets serve as the training dataset to adapt SAM to the medical domain, while *o.o.d.* datasets serve as unseen patient cases.

**Patient-Specific Segmentation Tasks.** We test P²SAM under two patient-specific segmentation tasks: NSCLC segmentation in the patient-adaptive radiation therapy and polyp segmentation in the endoscopy video. For NSCLC segmentation, medical image domain adaptation will be conducted on the *i.d.* dataset, NSCLC-Radiomics. For P²SAM, experiments are then carried out on the *o.o.d.* dataset, 4D-Lung. We evaluate P²SAM on patients who underwent multiple visits during treatment. For each patient, we utilize the image-mask pair from the first visit as the patient-specific prior data. For polyp segmentation, domain adaptation will be conducted on *i.d.* dataset, Kvasir-SEG. For P²SAM, experiments are then carried out on *o.o.d.* dataset, CVC-ClinicDB. For each video, we utilize the image-mask pair from the first stable frame as the patient-specific prior data.

**Implementation Details.** All experiments are conducted on A40 GPUs. For the NSCLC-Radiomics dataset, we extract 2-dimensional slices from the original computed tomography scans, resulting in a total of 7355 labeled images. As for the Kvasir-SEG dataset, we utilize all 1000 labeled images. We process two datasets following existing works (Hossain et al., 2019; Dumitru et al., 2023). Each dataset was randomly split into three subsets: training, validation, and testing, with an $80 : 10 : 10$ percent ratio (patient-wise splitting for the NSCLC-Radiomics dataset to prevent data leak). The model is initialized with the SAM's pre-trained weights and fine-tuned on the training splitting using the loss function proposed by SAM. We optimize the model by AdamW optimizer (Loshchilov & Hutter, 2017) ($\beta_1$=0.9, $\beta_2$=0.999), with a weight decay of 0.05. We further penalize the SAM's encoder with a drop path of 0.1. We fine-tune the model for 36 epochs on the NSCLC-Radiomics dataset and 100 epochs on the Kvasir-SEG dataset with a batch size of 4. The initial learning rate is $1e-4$, and the fine-tuning process is guided by cosine learning rate decay, with a linear learning rate warm-up over the first 10 percent epochs. More details are provided in Appendix C.

**Summary.** We test P²SAM on *o.o.d.* datasets with three different SAM backbones: 1. SAM pre-trained on the SA-1B dataset (Kirillov et al., 2023), denoted as *Meta*. 2. SAM adapted on *i.d.* datasets with LoRA (Hu et al., 2021) and 3. full fine-tune, denoted as *LoRA* and *Full-Fine-Tune*, respectively. We compare P²SAM against various methods, including previous approaches such as the *direct-transfer* baseline; *fine-tune* on the prior data (Wang et al., 2019a; Elmahdy et al., 2020; Wang et al., 2020; Chen et al., 2023); the one-shot segmentation method, PANet (Wang et al., 2019b); and concurrent methods that also utilize SAM, such as PerSAM (Zhang et al., 2023) and Matcher (Liu et al., 2023). For PANet, we utilize its align method for one-shot segmentation. For Matcher, we adopt its setting of FSS-1000 (Li et al., 2020). It is important to note that all baseline methods share the same backbone model as P²SAM does for fairness.

## 4.2 QUANTITATIVE RESULTS

**Patient-Adaptive Radiation Therapy.** As shown in Table 1, on the 4D-Lung dataset (Hugo et al., 2016), P²SAM outperforms all other baselines across various backbones. Notably, when utilizing *Meta*, P²SAM can outperform Matcher by +15.24% and PerSAM by +18.68% mean Dice score. This highlights P²SAM's superior adaptation to the out-of-domain medical applications. After domain adaptation, P²SAM can outperform the *direct-transfer* baseline by +8.01%, Matcher by +11.60%, and PerSAM by +2.48% mean Dice score. Demonstrate that P²SAM is a more effective method to enhance model generalization on the *o.o.d.* data.

Table 1: Results of NSCLC segmentation for patient-adaptive radiation therapy. We show the mean Dice score for each method. $base^{5.5M}$ indicates tuning $5.5M$ parameters of the base SAM on the NSCLC-Radiomics dataset before testing on the 4D-Lung dataset. † indicates training-free method; ‡ indicates the method using SAM.

| Method | Meta | LoRA | | Full-Fine-Tune | |
|---|---|---|---|---|---|
| | $huge^{0.0M}$ | $base^{5.5M}$ | $large^{5.9M}$ | $base^{93.8M}$ | $large^{312.5M}$ |
| *direct-transfer*† | - | 56.10 | 57.83 | 58.18 | 61.11 |
| *fine-tune* | - | 52.11 | 32.55 | 55.27 | 53.85 |
| PANet† (Wang et al., 2019b) | 4.28 | 5.24 | 7.79 | 40.03 | 44.70 |
| Matcher†‡ (Liu et al., 2023) | 13.28 | 50.81 | 50.88 | 59.52 | 57.67 |
| PerSAM†‡ (Zhang et al., 2023) | 9.84 | 63.63 | 64.69 | 62.58 | 64.45 |
| $P^2$SAM †‡ (Ours) | **28.52** | **64.38** | **67.00** | **66.68** | **67.23** |

Table 2: Results of polyp segmentation for endoscopy video. Similar to Table 1, we show the mean Dice score for each method. $base^{5.5M}$ indicates tuning $5.5M$ parameters of the base SAM on the Kvasir-SEG dataset before testing on the CVC-ClinicDB dataset.

| Method | Meta | LoRA | | Full-Fine-Tune | |
|---|---|---|---|---|---|
| | $huge^{0.0M}$ | $base^{5.5M}$ | $large^{5.9M}$ | $base^{93.8M}$ | $large^{312.5M}$ |
| *direct-transfer*† | - | 77.20 | 81.16 | 84.62 | 86.68 |
| *fine-tune* | - | 75.29 | 79.50 | 83.14 | 86.67 |
| PANet† (Wang et al., 2019b) | 38.22 | 44.61 | 55.48 | 75.99 | 86.48 |
| Matcher†‡ (Liu et al., 2023) | 63.54 | 78.65 | 79.56 | 85.17 | 87.15 |
| PerSAM†‡ (Zhang et al., 2023) | 45.82 | 79.02 | 81.63 | 85.74 | 87.88 |
| $P^2$SAM †‡ (Ours) | **66.45** | **80.03** | **82.60** | **86.40** | **88.76** |

**Discussion.** *fine-tune* is susceptible to overfitting with one-shot data, PANet fully depends on the encoder, and Matcher selects prompts based on patch-level features. These limitations prevent them from surpassing the *direct-transfer* baseline. On the other hand, NSCLC segmentation remains a challenging task. We consider MedSAM (Ma et al., 2024a), which has been pre-trained on a large-scale medical image dataset, as a strong *baseline* method. In Table 3, MedSAM achieves a 69% mean dice score on the 4D-Lung dataset with a human-given box prompt at each visit, while $P^2$SAM achieves comparable performance only with the ground truth provided at the first visit.

**Endoscopy Video.** As shown in Table 2, on the CVC-ClinicDB dataset (Bernal et al., 2015), $P^2$SAM still achieves the best result across various backbones. When utilizing *Meta*, $P^2$SAM can surpass Matcher by +2.91% and PerSAM by +20.63% mean Dice score. After domain adaptation, $P^2$SAM can outperform *direct-transfer* by +2.03%, Matcher by +1.81% and PerSAM by +0.88% mean Dice score. Demonstrates $P^2$SAM's generality to various patient-specific segmentation tasks.

**Discussion.** All methods demonstrate improved performance in datasets like CVC-ClinicDB, which exhibit a smaller domain gap (Matsoukas et al., 2022) with the SA-1B, SAM's pre-training dataset. In Table 3, we compare our results with Sanderson & Matuszewski (2022), which is reported as the method achieving the best performance in Dumitru et al. (2023) under the same evaluation objective: trained on Kvasir-SEG dataset and tested on the CVC-ClinicDB dataset. Our *direct-transfer* baseline has already surpassed this result, which can be attributed to the superior generality of SAM but our $P^2$SAM can further improve the generalization.

On the other hand, we observe that $P^2$SAM's improvements over PerSAM become marginal after domain adaptation (*LoRA* and *Full-Fine-Tune v.s. Meta*) on both datasets. This is because, as detailed in Appendix B, the ambiguity inherent in SAM, which is the primary limitation of PerSAM, is significantly reduced after fine-tuning on a dataset with a specific segmentation objective. Nevertheless, our method shows that providing multiple curated prompts can achieve further improvement.

Table 3: Comparison with existing base-lines. $\star$ indicates using a human-given box prompt during the inference time.

| Method | 4D-Lung | CVC-ClinicDB |
|---|---|---|
| *baseline* | $69.00^\star$ | 83.14 |
| *direct-transfer* | 61.11 | 86.68 |
| $P^2$SAM | 67.23 | 88.76 |

Table 4: Results of one-shot semantic segmentation. We show the mean IoU score for each method. Note that all methods utilize SAM's encoder for fairness.

| Method | COCO-$20^i$ | FSS-1000 | LVIS-$92^i$ | PerSeg |
|---|---|---|---|---|
| Matcher | 25.1 | 82.1 | 12.6 | 90.2 |
| PerSAM | 23.0 | 71.2 | 11.5 | 89.3 |
| $P^2$SAM (Ours) | **26.0** | **82.4** | **13.7** | **95.7** |

Table 5: Comparison with tracking methods. $*$ indicates utilizing *Full-Fine-Tune*.

| Method | 4D-Lung | CVC-ClinicDB |
|---|---|---|
| AOT | - | 62.34 |
| $P^2$SAM | - | 67.23 |
| SAM 2 | - | 81.98 |
| SAM 2 + $P^2$SAM | - | **84.43** |
| *label-propagation*$^*$ | 57.00 | 82.92 |
| $P^2$SAM $^*$ | **67.23** | **88.76** |

Table 6: Ablation study for the number of parts $n$ and the retrieval. Default settings are marked in Gray.

| # parts ($n$) | CVC-ClinicDB | | PerSeg | |
|---|---|---|---|---|
| | *w.o.* | *w.* retrieval | *w.o.* | *w.* retrieval |
| 1 (PerSAM) | 45.8 | 45.8 | 89.3 | 89.3 |
| 2 | 53.9 | 59.5 | 83.7 | 92.9 |
| 3 | 53.6 | 61.9 | 91.0 | 95.6 |
| 4 | 54.3 | 63.1 | 93.8 | 95.6 |
| 5 | 56.6 | 64.2 | 93.3 | 95.7 |

**Comparison with Tracking Algorithms.** In Table 5, we additionally compared $P^2$SAM with tracking algorithms: the *label-propagation* (Jabri et al., 2020) baseline, AOT (Yang et al., 2021), and SAM 2 (Ravi et al., 2024). On the 4D-Lung dataset, we only test algorithms with *Full-Fine-Tune* due to the large domain gap (Matsoukas et al., 2022). $P^2$SAM outperforms the *label-propagation* baseline, as the discontinuity in sequential visits—where the interval between two CT scans can exceed a week—leads to significant changes in tumor position and features. On the CVC-ClinicDB dataset, dramatic content shifts within the narrow field of view can also lead to discontinuity. Despite this, SAM 2 achieves competitive results even without additional domain adaptation. However, as we have stated, $P^2$SAM can be integrated into any promptable segmentation model. Indeed, we observe further improvements when applying $P^2$SAM to SAM 2.

**Existing One-shot Segmentation Benchmarks.** To further demonstrate $P^2$SAM can also be generalized to natural image domain, we evaluate its performance on existing one-shot semantic segmentation benchmarks: COCO-$20^i$ (Nguyen & Todorovic, 2019), FSS-1000 (Li et al., 2020), LVIS-$92^i$ (Liu et al., 2023), and a personalized segmentation benchmark, PerSeg (Zhang et al., 2023). We follow previous works (Zhang et al., 2023; Liu et al., 2023) for data pre-processing and evaluation. In Table 4, when utilizing SAM's encoder, $P^2$SAM outperforms concurrent works, Matcher and PerSAM, on all existing benchmarks. In addition, $P^2$SAM can achieve a new state-of-the-art result on the personalized segmentation benchmark, PerSeg (Zhang et al., 2023).

## 4.3 QUALITATIVE RESULTS

Figure 6 and 7 showcase the advantage of $P^2$SAM for out-of-domain applications. As shown in Figure 6, by presenting sufficient negative-point prompts, we enforce the model's focus on the semantic target. Results in Figure 7 further summarize the benefits of our method: unambiguous segmentation and robust prompts selection. Our $P^2$SAM can also improve the model's generalization after domain adaptation. By providing precise foreground information, $P^2$SAM enhances segmentation performance when the object is too small (*e.g.*, the first two columns in Figure 8) and when the segmentation is incomplete (*e.g.*, the last two columns in Figure 9). Figure 10 and 11 showcase the qualitative results on the PerSeg dataset, compared with Matcher and PerSAM respectively. The remarkable results demonstrate that $P^2$SAM can generalize well to different domain applications.

## 4.4 ABLATION STUDY

Ablation studies are conducted on the PerSeg dataset (Zhang et al., 2023) and CVC-ClinicDB dataset (Bernal et al., 2015) using *Meta*. We explore the effects of the number of parts in the part-aware prompt mechanism; the retrieval approach; distribution similarity measurements in the retrieval approach; and the model size, which can be considered a proxy for representation capacity.

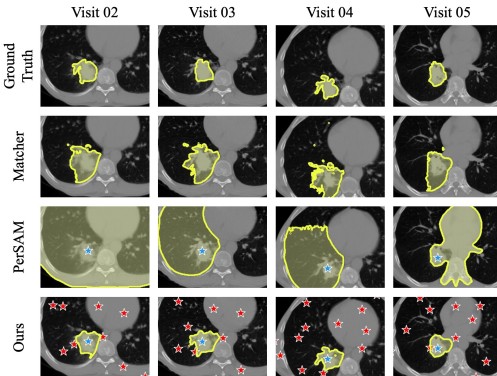

Figure 6: Qualitative results of NSCLC segmentation on the 4D-Lung dataset, with *Meta*.

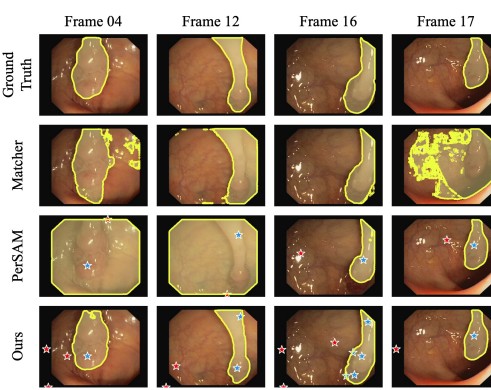

Figure 7: Qualitative results of polyp segmentation on the CVC-ClinicDB dataset, with *Meta*.

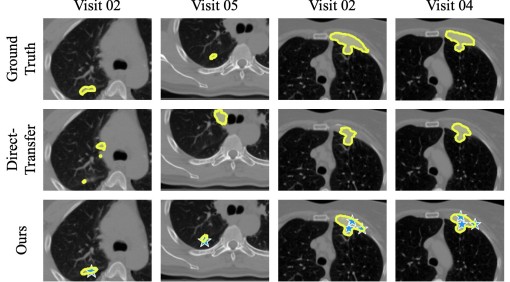

Figure 8: Qualitative results of NSCLC segmentation from two patients on the 4D-Lung dataset, with *Full-Fine-Tune*.

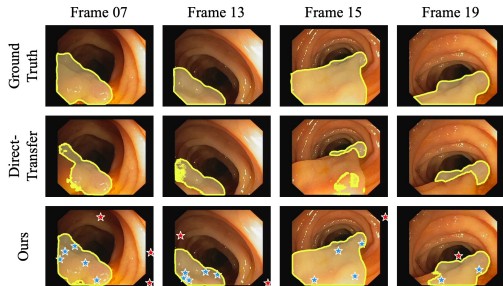

Figure 9: Qualitative results of polyp segmentation from one video on the CVC-ClinicDB dataset, with *Full-Fine-Tune*.

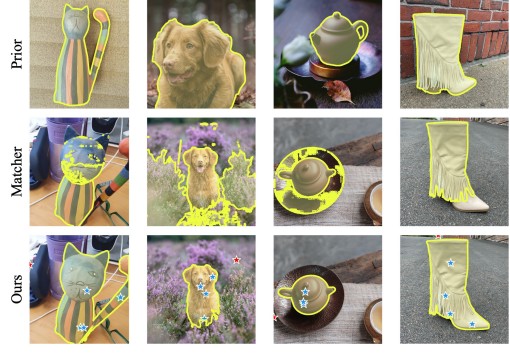

Figure 10: Qualitative results of personalized segmentation on the PerSeg dataset, compared with Matcher.

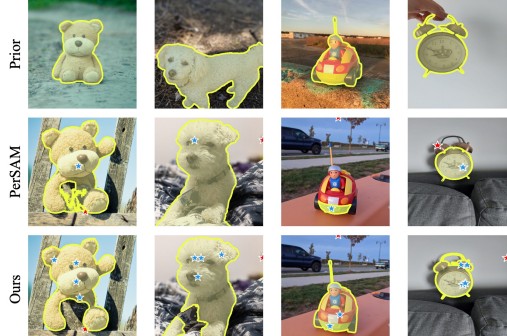

Figure 11: Qualitative results of personalized segmentation on the PerSeg dataset, compared with PerSAM.

**Number of Parts *n*.** To validate the efficacy of the part-aware prompt mechanism, we establish a method without the retrieval approach. As shown in Table 6 (*w.o.*), for both datasets, even solely relying on the part-aware prompt mechanism, increasing the number of parts $n$ enhances segmentation performance. When setting $n=5$, our part-aware prompt mechanism enhances performance by $+10.7\%$ mean Dice score on CVC-ClinicDB, $+4.0\%$ mean IoU score on PerSeg. These substantial improvements underscore the effectiveness of our part-aware prompt mechanism.

**Retrieval Approach.** The effectiveness of our retrieval approach is also shown in Table 6 (*w.* retrieval). When setting $n=5$, the retrieval approach enhances performance by $+7.6\%$ mean Dice score on the CVC-ClinicDB dataset, $+2.4\%$ mean IoU score on the PerSeg dataset. These substantial improvements show that our retrieval approach can retrieve an appropriate number of parts for different cases. Moreover, these suggest that we can initially define a wide range of part counts for retrieval, rather than tuning it meticulously as a hyperparameter.

Table 7: Ablation study for the distribution similarity measurement. Default settings are marked in Gray .

| Algorithm | CVC-ClinicDB | PerSeg |
|---|---|---|
| *w.o.* | 54.3 | 93.8 |
| *Hungarian* | 61.1 | 95.6 |
| *Jensen–Shannon* | 58.1 | 94.0 |
| *Wasserstein* | 63.1 | 95.6 |

Table 8: Ablation study for model sizes. ↑ indicates the improvement when compared with the same size PerSAM. Default settings are marked in Gray .

| Model | CVC-ClinicDB | PerSeg |
|---|---|---|
| $\text{PerSAM}^{huge}$ | 45.8 | 89.3 |
| $\text{P}^2\text{SAM}^{base}$ | 55.1 | 90.0 $_{26.0\uparrow}$ |
| $\text{P}^2\text{SAM}^{large}$ | 63.8 | 95.6 $_{9.0\uparrow}$ |
| $\text{P}^2\text{SAM}^{huge}$ | 63.1 | 95.6 $_{6.3\uparrow}$ |

**Distribution Similarity Measurements ($n$=4).** The cornerstone of our retrieval approach lies in distribution similarity measurements. To evaluate the efficacy of various algorithms, in Table 7, we juxtapose two distribution-related algorithms, namely *Wasserstein* distance (Rüschendorf, 1985) and *Jensen–Shannon* divergence (Menéndez et al., 1997), alongside a bipartite matching algorithm, *Hungarian* algorithm. Given foreground features from the reference image and the target image, we compute: 1. *Wasserstein* distance following the principles of WGAN (Arjovsky et al., 2017); 2. *Jensen-Shannon* divergence based on the first two principal components of each feature; 3. *Hungarian* algorithm after clustering these two sets of features into an equal number of groups. All algorithms exhibit improvements in segmentation performance compared to the *w.o.* retrieval baseline, while the *Wasserstein* distance is better in our context. Note that, the efficacy of the *Jensen-Shannon* divergence further corroborates our assumption that foreground features from the reference image and a correct target result should align in the same distribution, albeit it faces challenges when handling the high-dimensional data.

**Model Size ($n$=4).** In Table 8, we investigate the performance of different model sizes for our $\text{P}^2\text{SAM}$, *i.e.*, *base*, *large*, and *huge*, which can alternatively be viewed as the representation capacity of different backbones. For the CVC-ClinicDB dataset, a larger model size does not necessarily lead to better results. This result aligns with current conclusions (Mazurowski et al., 2023; Huang et al., 2024): In medical image analysis, the *huge* SAM may occasionally be outperformed by the *large* SAM. On the other hand, for the PerSeg dataset, even utilizing the *base* SAM, $\text{P}^2\text{SAM}$ achieves higher accuracy compared to PerSAM with the *huge* SAM. These findings further underscore the robustness of $\text{P}^2\text{SAM}$, particularly in scenarios where the model exhibits weaker representation, a circumstance more prevalent in medical image analysis.

## 5 CONCLUSION

We propose a data-efficient segmentation method, $\text{P}^2\text{SAM}$, to solve the patient-specific segmentation problem. With a novel part-aware prompt mechanism and a distribution-similarity-based retrieval approach, $\text{P}^2\text{SAM}$ can effectively integrate the patient-specific prior information into the current segmentation task. $\text{P}^2\text{SAM}$ demonstrates promising versatility in enhancing the backbone's generalization across various levels: 1. At the task level, $\text{P}^2\text{SAM}$ enhances performance across different patient-specific segmentation tasks. 2. At the model level, $\text{P}^2\text{SAM}$ can be integrated into various promptable segmentation models, such as SAM, SAM 2, and SAM after domain adaptation. 3. At the domain level, $\text{P}^2\text{SAM}$ performs effectively in both medical and natural image domains. We discuss a potential limitation of $\text{P}^2\text{SAM}$ in Appendix E. $\text{P}^2\text{SAM}$ may face challenges when multiple similar objects are present, a difficulty also encountered by other methods. While this scenario is uncommon in most patient-specific segmentation settings, we acknowledge this limitation and propose a potential solution. In this work, to meet clinical requirements, we choose to adapt SAM to the medical imaging domain with public datasets. We opted not to adopt SAM 2, as it requires video data for fine-tuning, which is more costly. Additionally, treating certain patient-specific segmentation tasks as video tracking is inappropriate. In contrast, approaching patient-specific segmentation as an in-context segmentation problem offers a more flexible solution for various patient-specific segmentation tasks. Moreover, $\text{P}^2\text{SAM}$ has demonstrated advantages when integrated with SAM 2 for polyp video segmentation even before domain adaptation, suggesting its potential to enhance performance in methods of segmenting medical video and in methods of segmenting 3D medical volumes as video. Further exploration of this potential is left for future work. We hope our work brings attention to the patient-specific segmentation problem within the research community.

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

APPENDIX

## A    SAM REVIEW

**Overview.** Segment Anything Model (SAM) (Kirillov et al., 2023) comprises three main components: an image encoder, a prompt encoder, and a mask decoder, denoted as $Enc_I$, $Enc_P$, and $Dec_M$. As a promptable segmentation model, SAM takes an image $I$ and a set of human-given prompts $P$ as input. SAM predicts segmentation masks $Ms$ by:

$$Ms = Dec_M(Enc_I(I), Enc_P(P)) \tag{3}$$

During training, SAM supervises the mask prediction with a linear combination of focal loss (Lin et al., 2017) and dice loss (Milletari et al., 2016) in a 20:1 ratio. When only a single prompt is provided, SAM generates multiple predicted masks. However, SAM backpropagates from the predicted mask with the lowest loss. Note that SAM returns only one predicted mask when presented with multiple prompts simultaneously.

**Prompt Encoder Details.** $Enc_I$ and $Dec_M$ primarily employ the Transformer (Vaswani, 2017; Dosovitskiy et al., 2020) architecture. Here, we provide details on components in $Enc_P$. $Enc_P$ supports three prompt modalities as input: the point, box, and mask logit. The positive- and negative-point prompts are represented by two learnable embeddings, denoted as $E_{\texttt{pos}}$ and $E_{\texttt{neg}}$, respectively. The box prompt comprises two learnable embeddings representing the left-up and right-down corners of the box, denoted as $E_{\texttt{up}}$ and $E_{\texttt{down}}$. In cases where neither the point nor box prompt is provided, another learnable embedding $E_{\texttt{not-a-point}}$ is utilized. If available, the mask prompt is encoded by a stack of convolution layers, denoted as $E_{\texttt{mask}}$; otherwise, it is represented by a learnable embedding $E_{\texttt{not-a-mask}}$.

**Interactive Training.** SAM employs an interactive training strategy. In the first iteration, either a positive-point prompt, represented by $E_{\texttt{pos}}$, or a box prompt, represented by $\{E_{\texttt{up}}, E_{\texttt{down}}\}$, is randomly selected with equal probability from the ground truth mask. Since there is no mask prompt in the first iteration, $E_{\texttt{pos}}$ or $\{E_{\texttt{up}}, E_{\texttt{down}}\}$ is combined with $E_{\texttt{not-a-mask}}$ and fed into $Dec_M$. In the follow-up iterations, subsequent positive- and negative-point prompts are uniformly selected from the error region between the predicted mask and the ground truth mask. SAM additionally provides the mask logit prediction from the previous iteration as a supplement prompt. As a result, $\{E_{\texttt{pos}}, E_{\texttt{neg}}, E_{\texttt{mask}}\}$ is fed into $Dec_M$ during each iteration. There are 11 total iterations: one sampled initial input prompt, 8 iteratively sampled points, and two iterations where only the mask prediction from the previous iteration is supplied to the model.

## B    SAM ADAPTATION DETAILS

In Section 3.3, we propose to adapt SAM to the medical image domain when it is needed, with full fine-tune (*Full-Fine-Tune*) and LoRA (Hu et al., 2021) (*LoRA*). For *Full-Fine-Tune*, we fine-tune all parameters in SAM backbone. For *LoRA*, we insert the LoRA module in the image encoder $Enc_I$ and only fine-tune parameters in the LoRA module and the mask decoder $Dec_M$. Our fine-tuning objectives are as follows:

1. The model can accurately predict a mask even if no prompt is provided.
2. The model can predict an exact mask even if only one prompt is given.
3. The model maintains promptable ability.

The training strategy outlined in SAM cannot satisfy all these three requirements: 1. The mask decoder $Dec_M$ is not trained to handle scenarios where no prompt is given. 2. The approach to resolving the ambiguous prompt by generating multiple results is redundant as we have a well-defined segmentation objective. Despite that, we find a simple modification can meet all our needs:

1. In the initial iteration, we introduce a scenario where no prompt is provided to SAM. As a result, $\{E_{\texttt{not-a-point}}, E_{\texttt{not-a-mask}}\}$ is fed into $Dec_M$ in the first iteration.

2. To prevent $E_{\texttt{not-a-point}}$ and $E_{\texttt{not-a-mask}}$ from introducing noise when human-given prompts are available, we stop their gradients in every iteration.

3. We ensure that SAM always returns an exact predicted mask. As a result, the ambiguity problem does not exist in the model after fine-tuning.

## C   TEST IMPLEMENTATION DETAILS

In this section, for further reproducibility, we provide the details of the retrieval range during the test time for the COCO-$20^i$ (Nguyen & Todorovic, 2019), FSS-1000 (Li et al., 2020), LVIS-$92^i$ (Liu et al., 2023), and Perseg (Zhang et al., 2023) dataset in Table 9, the 4D-Lung (Hugo et al., 2016) and CVC-ClinicDB (Bernal et al., 2015) dataset in Table 10,.

Table 9: Retrieval range for the COCO-$20^i$, FSS-1000, LVIS-$92^i$, PerSeg dataset. Blue indicates the retrieval range for positive-point prompts. Red indicates the retrieval range for negative-point prompts.

| COCO-$20^i$ | FSS-1000 | LVIS-$92^i$ | PerSeg |
|---|---|---|---|
| $1, 6{-}10$ / $1$ | $1{-}5$ / $1$ | $1, 4{-}10$ / $1$ | $1{-}5$ / $1$ |

Table 10: Retrieval range for the 4D-Lung and CVC-ClinicDB dataset. Blue indicates the retrieval range for positive-point prompts. Red indicates the retrieval range for negative-point prompts.

| Dataset | *Meta* | *LoRA* | | *Full-Fine-Tune* | |
|---|---|---|---|---|---|
| | *huge* | *base* | *large* | *base* | *large* |
| 4D-Lung | $1{-}2$ / $45$ | $1{-}3$ / $1$ | $1{-}3$ / $1$ | $1{-}3$ / $1$ | $1{-}3$ / $1$ |
| CVC-ClinicDB | $1{-}5$ / $1{-}3$ | $1{-}3$ / $1{-}3$ | $1{-}2$ / $1{-}3$ | $1{-}2$ / $1$ | $1{-}5$ / $1{-}3$ |

The final number of positive-point and negative-point prompts is determined by our distribution-similarity-based retrieval approach. Below, we explain how the retrieval range is determined.

For *LoRA* and *Full-Fine-Tune*, the retrieval range is determined based on the validation set of the *i.d.* datasets. We uniformly sample positive-point and negative-point prompts on the ground-truth mask and perform interactive segmentation. The number of prompts is increased until the improvement becomes marginal, at which point this maximum number is set as the retrieval range for *o.o.d.* test datasets. On the 4D-Lung dataset, we consistently set the number of negative-point prompts to 1 for these two types of models. This decision is informed by conclusions from previous works (Ma et al., 2024a; Huang et al., 2024), which suggest that the background and semantic target can appear very similar in CT images, and using too many negative-point prompts may confuse the model.

On the CVC-ClinicDB dataset, the endoscopy video is in RGB space, resulting in a relatively small domain gap (Matsoukas et al., 2022) compared to SAM's pre-trained dataset. Therefore, for *Meta*, we use the same retrieval range as the *Full-Fine-Tune* large model. In contrast, on the 4D-Lung dataset, CT images are in grayscale, leading to a significant domain gap (Matsoukas et al., 2022) compared to SAM's pre-trained dataset. Consequently, we set the retrieval range for positive-point prompts to 2 to avoid outliers and fixed the number of negative-point prompts to a large constant (*i.e.*, 45) rather than a range, to ensure the model focuses on the semantic target. These values were not further tuned.

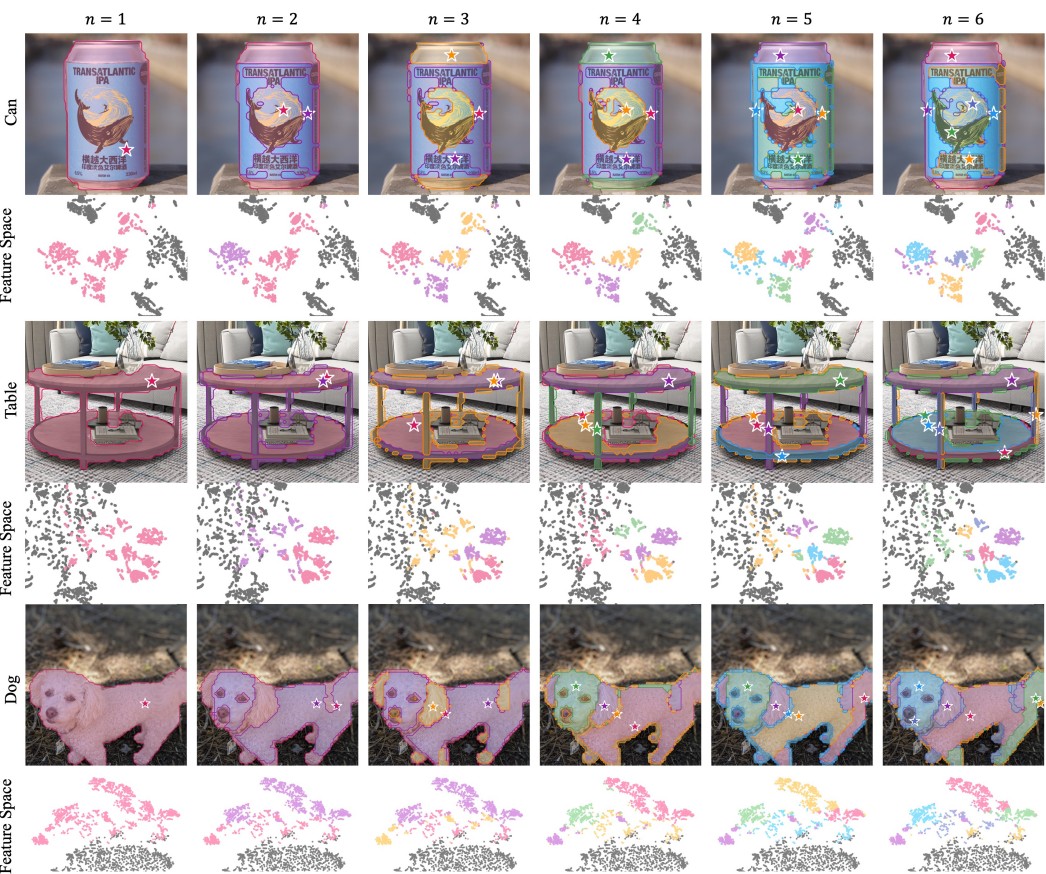

Figure 12: Visualization results on the 4D-Lung dataset, based on a varying number of part-level features.

Figure 13: Visualization results on the CVC-ClinicDB dataset, based on a varying number of part-level features.

Figure 14: Visualization results on the PerSeg dataset, based on a varying number of part-level features.

## D VISUALIZATION

In this section, to provide deeper insight into our part-aware prompt mechanism and distribution-similarity-based retrieval approach, we present additional visualization results on the 4D-Lung (Hugo et al., 2016) dataset, the CVC-ClinicDB (Bernal et al., 2015) dataset, and the PerSeg (Zhang et al., 2023) dataset. These visualizations are based on a varying number of part-level features, offering a clearer understanding of how the part-aware prompt mechanism adapts to different segmentation tasks and domains.

In Figure 12 and 13, we observe that an appropriate number of part-level features can effectively divide the tumor into distinct parts, such as the body and edges for non-small cell lung cancer, and

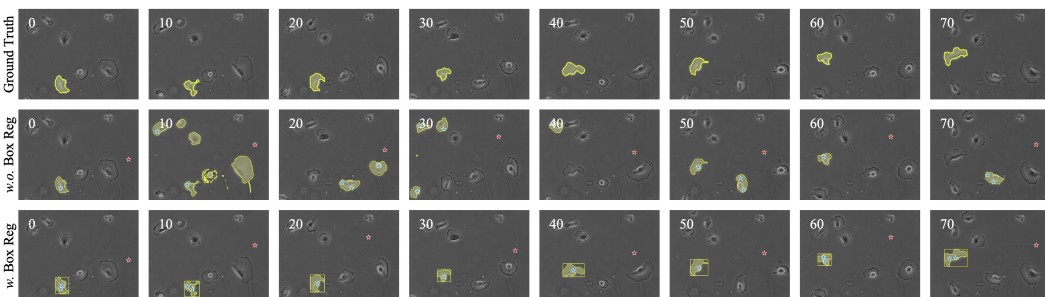

Figure 15: Qualitative results of single-cell segmentation on the PhC-C2DH-U373 dataset. The second row highlights the challenge P$^2$SAM faces in handling multiple similar objects. The third row demonstrates that P$^2$SAM can overcome this challenge with a cost-free regularization.

the body and light point (caused by the camera) for the polyp. This illustrates how P$^2$SAM can assist in cases of incomplete segmentation. In Figure 14, we observe that an appropriate number of part-level features can effectively divide the object into meaningful components, such as the pictures, characters, and aluminum material of a can; the legs and platforms of a table; or the face, ears, and body of a dog. These parts can merge naturally based on texture features when using the appropriate number of part-level features, whereas using too many features may result in over-segmentation. Our retrieval approach, on the other hand, helps determine the optimal number of part-level features for each specific case.

## E    MULTIPLE OBJECTS

In this section, we want to discuss a potential limitation of P$^2$SAM. P$^2$SAM demonstrates improvements in the backbone's generalization across domain, task, and model levels. At the task level, we have already shown how P$^2$SAM enhances performance for NSCLC segmentation in patient-adaptive radiation therapy and polyp segmentation in endoscopy videos. However, when addressing specific tasks that involve multiple similar targets, P$^2$SAM may fail. Although this scenario is uncommon in patient-specific segmentation, we acknowledge that P$^2$SAM faces the same challenge of handling multiple similar objects as other methods (Zhang et al., 2023; Liu et al., 2023). In Figure 15, we present an example of single-cell segmentation on the PhC-C2DH-U373 dataset (Maška et al., 2014), which goes beyond the patient-specific setting. In Figure 15, the second row illustrates that P$^2$SAM fails to segment the target cell due to the presence of many similar cells in the field of view. However, given the slow movement of the cell, we can leverage its previous information to regularize the current part-aware prompt mechanism. The third row in Figure 15 demonstrates that when using the bounding box from the last frame, originally propagated from the reference frame, to regularize the part-aware prompt mechanism in the current frame, P$^2$SAM achieves strong performance on the same task. Since the bounding box for the first frame can be generated from the ground truth mask, which is already available, this regularization incurs no additional cost. Utilizing such tailored regularization incorporating various prompt modalities, we showcase our approach's flexible applicability to other applications.

## F    DISCUSSION ON ADDITIONAL RELATED WORKS

**Interactive Segmentation for Medical Images.** Complementary to the works discussed in Section 2, several studies (Butoi et al., 2023; Wong et al., 2023; Ma et al., 2024a;b; Wu & Xu, 2024) have aimed to develop promptable segmentation models specifically for medical image segmentation. UniverSeg (Butoi et al., 2023) utilizes a support set to provide additional information to the model during the test time. In this work, we did not include UniverSeg as a baseline method because our problem setting provides only a single image-mask pair, and UniverSeg's performance significantly declines under such conditions. Moreover, UniverSeg employs a different backbone model and training objective, making it challenging to test on our dataset. Other methods, such as ScribblePrompt (Wong et al., 2023), One-Prompt (Wu & Xu, 2024), and MedSAM2 (Ma et al., 2024b), primarily focus on interactively segmenting medical images. In contrast, our work presents

an effective approach that leverages patient-specific prior data to address segmentation for out-of-distribution patient samples that lie outside the training distribution. Among them, we have chosen MedSAM Ma et al. (2024a) for comparison in Table 3, as it was pre-trained on a large-scale medical image dataset and supplemented with a human-given box prompt during inference. It is worth noting that other methods either utilized smaller pre-training datasets or were not available at the time this work was conducted. On the other hand, utilizing other prompt modalities, such as scribble, mask, and box, presents challenges for solving the patient-specific segmentation problem, as it is difficult to represent prior data in these formats. In this work, we adopt a more flexible prompt modality: point prompts. Although it may be possible to convert our multiple-point prompts into a scribble prompt by connecting them together, we leave the exploration of this direction for future work.

**MedSAM as a Strong Baseline.** At the outset, we would like to clarify that this paper focuses on the task of external validation (*o.o.d.*), where the testing dataset differs from the distribution of the training dataset. In this scenario, the model's generalization ability becomes critical for achieving better performance. We acknowledge that MedSAM is widely used as a baseline across many benchmarks (Antonelli et al., 2022; Ji et al., 2022). However, these comparisons primarily focus on internal (*i.d.*) validation. MedSAM has the potential to outperform many models on external validation sets due to its pre-training on a large-scale medical image dataset. While there is no direct evidence to confirm this, DuckNet (Dumitru et al., 2023) (Table 1 *v.s.* Table 5) suggests that large-scale pre-trained models generally outperform others on external validation sets, even if they lag behind on internal validation. The 4D-Lung dataset (Hugo et al., 2016) is a relatively new benchmark for longitudinal data analysis, and no standard benchmark for comparison was available at the time this work was conducted. The results from MedSAM in Table 3 could serve as a strong baseline, particularly when supplemented with human-provided box prompts. Therefore, we consider MedSAM a reliable baseline for comparison, especially for external validation, given its generalization ability.

**Different SAM Adaptation Methods.** The main purpose of P$^2$SAM is to leverage one-shot patient-specific prior data to address segmentation for *o.o.d.* patient samples. Under this objective, the SAM adaptation is an optional and orthogonal procedure that can be employed when limited labeled data is available to further enhance the pre-trained model. In this work, we test full fine-tune method and LoRA (Hu et al., 2021) for parameter-efficient fine-tuning. When compared with other parameter-efficient fine-tuning strategies like Adapter (Houlsby et al., 2019) and Prompt-Tuning Li & Liang (2021), LoRA integrates the learned parameters directly into the original model, ensuring no additional latency during inference. Since P$^2$SAM can be integrated with any backbone model that supports the point-prompt modality, it is compatible with various parameter-efficient fine-tuning methods, such as Adapter or Prompt-Tuning, as adopted in Med-SA (Wu et al., 2023).

