# OpenReview forum: "Part-aware Personalized Segment Anything Model for Patient-Specific Segmentation"
_ICLR.cc/2025/Conference — Submitted to ICLR 2025_

### Official Review · Reviewer_9LGG · 2024-10-26

**Soundness:** 2
**Presentation:** 3
**Contribution:** 2
**Rating:** 8
**Confidence:** 5

**Summary:**

This paper introduces P2SAM, a novel data-efficient segmentation approach tailored for precision medicine, where patient-specific variability and limited annotated data challenge traditional segmentation models. Unlike conventional methods that require fine-tuning, P2SAM achieves adaptation to new patients using only one-shot patient-specific data.

**Strengths:**

1. the task in significant in practice
2. the paper is easy to follow

**Weaknesses:**

1. my main concern is the paper ignores many significant previous work[1,2,3] doing the same task (segmenting follow-up medical images given a template image with prompt or mask). Many of previous works have been explored this task with much larger scale training, more comprehensive experimental testing, and human evaluation. The paper is encouraged to compare with them or highlight the difference with them.
2. the paper takes MedSAM as a SOTA in its comparisons, however, MedSAM serves basically as a baseline and does not represent SOTA on most tasks.
3. the paper used adapter to efficiently fine-tune SAM on medical image segmentation while it is not compared with Med-SAM-Adapter[4] as a baseline.
4. the paper is also encouraged do comparison under diverse visual prompts beyond point and bbox, like scribble[5, 2] and mask[1], which have been widely-used and comprehensively explored in the medical image segmentation before.

1. Butoi, Victor Ion, et al. "Universeg: Universal medical image segmentation." Proceedings of the IEEE/CVF International Conference on Computer Vision. 2023.
2. Wu, Junde, and Min Xu. "One-prompt to segment all medical images." Proceedings of the IEEE/CVF Conference on Computer Vision and Pattern Recognition. 2024.
3. Zhu, Jiayuan, et al. "Medical sam 2: Segment medical images as video via segment anything model 2." arXiv preprint arXiv:2408.00874 (2024).
4. Wu, Junde, et al. "Medical sam adapter: Adapting segment anything model for medical image segmentation." arXiv preprint arXiv:2304.12620 (2023).
5. Wong, Hallee E., et al. "Scribbleprompt: Fast and flexible interactive segmentation for any medical image." arXiv preprint arXiv:2312.07381 (2023).

**Questions:**

see weakness

---

> ### Author Response · Authors · 2024-11-18
> **Response to reviewer 9LGG**
>
> We thank you for your comments and the approval of our target task. In addition to the general response, we address your concerns below:
>
> ### W1: Ignores many significant previous works
>
> - UniverSeg employs a different backbone model and training objective, making it difficult to test on our dataset. Additionally, UniverSeg's performance significantly declines when its support set contains only a single image-mask pair.
>
> - Other methods, such as MedSAM, MedSAM2, One-Prompt, and ScribblePrompt, primarily focus on interactively segmenting medical images. In contrast, our work introduces an effective approach that leverages patient-specific prior data to address segmentation for out-of-distribution patient samples, which fall outside the training distribution.
>
> - We have chosen MedSAM for comparison in Table 3, as it was pre-trained on a large-scale medical image dataset and supplemented with a human-given box prompt during inference. Note that other methods either used smaller pre-training datasets or were not available at the time this work was conducted.
>
> - Consequently, the most relevant baseline methods remain PerSAM and Matcher.
>
> Nonetheless, thank you for pointing this out. We have now included these works in the related work section to discuss these concurrent approaches.
>
> ---
>
> ### W2: The paper takes MedSAM as a SOTA.
>
> - Firstly, we would like to clarify that this paper focuses on the task of external validation (out-of-distribution dataset), where the testing dataset is not from the same distribution as the training dataset. In this scenario, the model’s generalization ability becomes crucial for achieving better performance.
>
> - We acknowledge that MedSAM is widely used as a baseline across many benchmarks; however, these comparisons often focus on internal validation sets (in-distribution dataset). MedSAM has the potential to outperform many models on external validation sets given its pre-training on a large-scale medical image dataset. While there is no direct evidence to confirm this, DuckNet [1] (Table 1 vs Table 5 ) demonstrates that large-scale pre-trained models generally outperform others on external datasets, even if they lag behind on internal validation sets.
>
> - The 4D-Lung dataset is a relatively new benchmark for longitudinal data analysis, and no standard benchmark for comparison was available at the time this work was conducted. The results from MedSAM in Table 3 demonstrate that it serves as a strong baseline, particularly when supplemented with human-provided box prompts. Therefore, we consider MedSAM a reliable baseline for comparison, especially on external validation sets given its generalization ability. To avoid confusion, we will revise the text in Table 3 to clarify this.
>
> [1] Dumitru, Razvan-Gabriel, Darius Peteleaza, and Catalin Craciun. "Using DUCK-Net for polyp image segmentation." Scientific reports 13.1 (2023): 9803.
>
> ---
>
> ### W3: Comparison under diverse visual prompts beyond the point prompt.
>
> The main challenge in utilizing other prompt modalities, such as scribble, mask, and box, lies in how to represent the prior data in these formats. In this work, we adopt a more flexible prompt modality: point prompts. While it may be possible to convert our multiple-point prompts into a scribble prompt, we leave the exploration of this direction for future work.
>
> ---
>
> ### W4: Not compared with Med-SAM-Adapter as a baseline.
>
> - Our work introduces an effective approach that leverages one-shot patient-specific prior data to address segmentation for out-of-distribution patient samples. The fine-tuning stage is an optional module that can be utilized when limited labeled data is available to further enhance the pre-trained model.
>
> - In this work, we choose Low-Rank Adaptation (LoRA) over Adapters. While Adapters add extra parameters during the inference stage, potentially increasing latency and slowing down inference, LoRA integrates the learned parameters directly into the original model, ensuring no additional latency during inference.
>
> - Our proposed method is compatible with various model fine-tuning strategies to further improve performance. As such, it is orthogonal to other approaches, such as adapter methods.
>
> ---

---

> > ### Comment · Reviewer_9LGG · 2024-11-18
> > **response**
> >
> > agree on most points.
> >
> > Med-SAM-Adapter can now easily switch to LoRA in their code: https://github.com/MedicineToken/Medical-SAM-Adapter, I still encourage to integrate a comparison or discussion.
> >
> > would like to improve rate if authors can reflect all those mentioned in their revision.

---

> > > ### Author Response · Authors · 2024-11-19
> > > **Response to reviewer 9LGG (Part 2)**
> > >
> > > Thank you for agreeing with our arguments in the last response. All modifications have been highlighted in blue in the uploaded revised manuscript. For your inference, just to list the key changes as below:
> > > 1. We have added a new section, Appendix F, to discuss these additional related works.
> > > 2. Furthermore, we have incorporated these related works and citations into Section 2 (Related Works).
> > > 3. In Section 4 (Experiments), we clarified that MedSAM is set as a strong baseline rather than the SOTA method, to avoid confusion.
> > >
> > > Regarding the question about Med-SA, to the best of our knowledge, Med-SA and our work share a similar training strategy. A subtle difference lies in our introduction of a scenario where no prompt is provided to the model at the initial iteration, as detailed in Appendix B. Upon reviewing the implementation of Med-SA, we found that its use of LoRA is very similar to the strategy employed in this work. Therefore, we believe our results closely align with those of Med-SA. In this case, we have discussed the compatibility of our method with Med-SA in Appendix F and further highlighted Med-SA in Section 3 (Method).

---

### Official Review · Reviewer_3qfj · 2024-11-02

**Soundness:** 3
**Presentation:** 3
**Contribution:** 4
**Rating:** 5
**Confidence:** 5

**Summary:**

This study introduces a segmentation model called P2SAM which enables efficient, personalized medical image segmentation based on one-shot patient-specific data, improving segmentation performance across different patients and domains.

**Strengths:**

1. The work introduces a "part-aware prompt mechanism" that leverages one-shot, patient-specific data for segmentation without additional fine-tuning. This feature is particularly suited to precision medicine applications where data annotation is minimal.

2. Unlike standard SAM applications, P2SAM integrates a distribution-similarity-based retrieval approach that optimizes the selection of prompts, enhancing segmentation performance and reducing ambiguities.

3. P2SAM is demonstrated to be effective across a variety of medical and natural image datasets, making it versatile for both patient-specific and general segmentation tasks.

4. Quantitative Improvements: The model shows notable improvement over existing methods (PerSAM and Matcher), with significant gains in Dice scores and mIoU in various datasets, proving the robustness of the approach in handling out-of-distribution medical images.

5. The paper presents extensive comparisons across multiple datasets and includes qualitative results that illustrate the model’s adaptability and performance in challenging segmentation tasks.

**Weaknesses:**

1. Complexity and Computational Cost: The proposed method, while innovative, involves complex modules (e.g., retrieval approach for optimal prompt selection) that may increase computational demands, potentially impacting usability in real-time clinical applications.

2. Limited Generalization to Other Modalities: Although versatile within specific settings, the method's performance in more diverse medical imaging modalities (e.g., MRI or ultrasound) remains unexplored.

3. Ambiguity in Module Contributions: The study could benefit from additional ablation studies to further clarify each component's role, such as the part-aware prompt mechanism versus the distribution-similarity-based retrieval.

4. Dependence on SAM Backbone: While P2SAM leverages SAM effectively, it is still heavily reliant on SAM’s architecture. Any significant updates to SAM or its successors may require substantial adaptation of P2SAM.

**Questions:**

1. Scalability to Other Imaging Tasks: Could P2SAM be effectively adapted to other domains, such as pathological imaging, or expanded to handle 3D volumetric data with higher efficiency?

2. Integration with SAM 2: The authors briefly mention SAM 2. How would the proposed retrieval mechanism and prompt system adapt to SAM 2's expanded capabilities in video and sequential data processing?

---

> ### Author Response · Authors · 2024-11-18
> **Response to reviewer 3qfj**
>
> We thank you for your comments and the approval of our target task. In addition to the general response, we address your concerns below:
>
> ### W1: Complexity and Computational Cost.
>
> On a cluster with 4 CPU cores, 32GB of CPU memory, and 1 A40 GPU, we observe the following computational times: K-Means clustering takes 0.1 seconds, computing the Wasserstein Distance takes 0.2 seconds, and model inference takes 0.5 seconds. The search ranges are provided in Tables 9 and 10 in APPENDIX C. Thus, the computational cost of the part-aware prompt mechanism and the retrieval approach is comparable to the cost of the model itself. Moreover, our method does not require any GPU fine-tuning. To enable real-time applications, the speed can be further enhanced through multi-session computation.
>
> ---
>
> ### W2: Limited Generalization to Other Modalities.
>
> Segmentation for longitudinal data remains an underexplored area in the community. In this work, we select the two most standard and widely used benchmarks, 4D-Lung and CVC-ClinicDB, which are commonly referenced in related works. On the other hand, extensive experiments have demonstrated that our method effectively addresses two distinct patient-specific segmentation tasks across various medical imaging modalities: NSCLC segmentation for adaptive radiation therapy using CT scans and polyp segmentation in endoscopy videos. Additionally, we have validated the effectiveness of our method on larger and more general natural image datasets. Therefore, we believe that our method can be seamlessly integrated into other imaging modalities like MRI or Ultrasound and applied to a wide range of diseases.
>
> ---
>
> ### W3: Ambiguity in Module Contributions
>
> In Table 6, we separate the part-aware prompt mechanism from the distribution-similarity-based retrieval approach. Results on both the CVC-ClinicDB and PerSeg datasets show that the part-aware prompt mechanism alone improves performance, with the retrieval approach providing additional enhancements.
>
> ---
>
> ### W4: Dependence on SAM Backbone
>
> - Our method can be integrated into any promptable segmentation model that supports the point-prompt modality. The two main modules of our method—the part-aware prompt mechanism and the retrieval approach—require only the image encoder from the backbone model as input and output the curated multiple-point prompts.
>
> - In this work, we chose SAM due to its pre-training on large-scale datasets. As demonstrated in Table 5, our method integrates seamlessly with SAM2, enhancing its performance on polyp segmentation. Consequently, we believe that our method can be effectively adapted to future successors of SAM.
>
> ---
>
> ### Q1: Limited Generalization to 3D Data
>
> We utilize a 2D model for the following reasons: 1. A 2D model is versatile and can accommodate various medical imaging modalities, such as endoscopy videos and NSCLC CT scans. 2. A 2D model effectively leverages the SAM pre-trained model, significantly enhancing model generalization. This approach also aligns with the strategy employed by MedSAM. Moreover, as we have discussed in the **DISCUSSION SECTION**, our method can also integrate with SAM2. This flexibility enables our method to efficiently complement several concurrent works [1] utilizing SAM2 for processing 3D medical image volumes. Finally, when powerful 3D pre-trained backbones become available, our model can be readily adapted to them.
>
> [1] Zhu, Jiayuan, et al. "Medical sam 2: Segment medical images as video via segment anything model 2." arXiv preprint arXiv:2408.00874 (2024).
>
> ---
>
> ### Q2: Integration with SAM 2
>
> When processing sequential data, SAM2 relies on a memory bank to incorporate information from previous data. Additionally, SAM2 can utilize optional human-provided prompts for each follow-up data point. When integrating our method with SAM2, we first apply our approach to generate multiple-point prompts for each follow-up data point. These prompts are then provided to SAM2’s mask decoder, which leverages both the multiple-point prompts and the memory bank to enhance segmentation performance on the follow-up data.
>
> ---

---

> > ### Comment · Reviewer_3qfj · 2024-11-22
> >
> > Thank you for your response. Based on the feedback and responses from other reviewers, I have decided to maintain this score.

---

> > > ### Author Response · Authors · 2024-11-22
> > > **Response to reviewer 3qfj (Part 2)**
> > >
> > > We sincerely thank you for your great efforts in reviewing this paper. We have gone through your points one by one and tried to address them carefully. Please don’t hesitate to let us know if you have any further questions.

---

### Official Review · Reviewer_mmQa · 2024-11-03

**Soundness:** 3
**Presentation:** 3
**Contribution:** 3
**Rating:** 5
**Confidence:** 4

**Summary:**

The authors target patient-specific segmentation by proposing a new segmentation pipeline: (1) obtaining multiple point-prompts from the part-aware prompt mechanism, and (2) feeding these point-prompts to a prompt-based segmentation network, such as SAM. The authors also propose a similarity-based refinement to control the number of prompts found during step 1. Main experiments were conducted on the NSCLC and CVC-ClinicDB datasets where the proposed method achieves the state-of-the-art performances. Further experiments were conducted to show the method's performance against tracking algorithms and on the one-shot segmentation task.

**Strengths:**

1. The paper is well-written and easy to follow.
2. The experiments are thorough, covering various important and related topics.
3. The proposed refinement strategy on the number of prompts is conceptually easy and seems effective.

**Weaknesses:**

1. Unclearness in writing: (a) The motivation for Table 3 is unclear. It overlaps with Table 1 and 2 and brings little additional information. (b) For the baseline methods: direct-transfer and fine-tune, looks like no prompts are provided during the evaluation stage. Can the authors verify this is true?
2. The contribution on the Part-aware Prompt Mechanism is unclear. Although the authors have demonstrated the effectiveness of the overall method, it is unclear if the improvements are from the part-based sampling strategy or merely because of having multiple prompts.
3. (Minor) The application range of the method is limited to patients with multi-exams.

**Questions:**

1. To demonstrate the effectiveness of the Part-aware Prompt Mechanism, could the author compared to a modified version of PerSAM where instead of selecting "two points with the highest and lowest confidence values", the authors may select the top K and bot K' points where K and K' matches P^2SAM. In this way, the authors can demonstrate that points found by the Part-aware Prompt Mechanism are more effective.
2. Since authors have shown better one-shot segmentation performance, I am curious to see if the authors can show their method's performance beyond patients with multi-exams. For example, for some medical segmentation datasets without patients with multi-exams, can the author randomly select one image-mask pair as the prior and test the model's performance on other pairs.

---

> ### Author Response · Authors · 2024-11-18
> **Response to reviewer mmQa**
>
> We thank you for your comments and the approval of our target task. In addition to the general response, we address your concerns below:
>
> ### W1: Unclearness in writing: Table 3 and two baseline methods.
>
> - The purpose of Table 3 is to provide the context of existing results to readers who may not be familiar with the CVC-ClinicDB and 4D-Lung benchmarks. Notably, the 4D-Lung dataset is a relatively new benchmark for longitudinal data analysis, and no standard benchmark was set up for comparison. Therefore, we designate the result from MedSAM, supplemented with human-provided box prompts, as a strong baseline result on this dataset. To avoid confusion, we will revise the text in Table 3 to clarify this.
>
> - The baseline methods, **direct-transfer** and **fine-tune**, do not rely on any human-provided prompts. The **fine-tune** method, previously employed by related works [1] for patient-specific segmentation, serves as a strong baseline. The **direct-transfer** method is considered a fair baseline for the following reasons:
>   - Before the evaluation stage, we fine-tune SAM on in-distribution datasets, as detailed in Appendix B. Importantly, our fine-tuned SAM can segment input images without requiring human-provided prompts due to modifications in the fine-tuning objective.
>   - Leveraging SAM's large-scale pre-training, the fine-tuned model achieves high performance even on out-of-distribution datasets. For instance, as shown in Table 3, the **direct-transfer** method outperforms existing methods on the CVC-ClinicDB dataset.
>   - When prior data is not effectively utilized, some methods, such as Matcher, may fail to surpass the **direct-transfer** baseline.
>
> - Additionally, we have compared our method with other more relevant baseline methods, including PerSAM and Matcher.
>
> [1] Yizheng Chen, Michael F Gensheimer, Hilary P Bagshaw, Santino Butler, Lequan Yu, Yuyin Zhou, Liyue Shen, Nataliya Kovalchuk, Murat Surucu, Daniel T Chang, et al. Patient-specific auto-segmentation on daily kvct images for adaptive radiotherapy. International Journal of Radiation Oncology* Biology* Physics, 2023.
>
> ---
>
> ### W2 and Q1: The contribution of the Part-aware Prompt Mechanism is unclear.
>
> - First, the contribution of the part-aware prompt mechanism has been demonstrated in Tables 1 and 2, where it outperforms Matcher. The results for Matcher indicate that when multiple prompts are not carefully selected, performance can occasionally fall below the direct-transfer baseline.
>
> - The contribution of the part-aware prompt mechanism is further supported by the experimental results on PerSeg. When modifying PerSAM with K positive-point prompts and K′ negative-point prompts, the results are shown as follows. The decrease is because: 1. Top-k-point prompts are often clustered closely together, providing limited additional information to the model. 2. Top-k-point prompts potentially increasing outlier prompts. In contrast, the part-aware prompt mechanism generates point prompts based on each distinct part of the target, resulting in more diverse, informative, and robust prompts.
>
> | $P^2SAM$ (K=5, K'=1) | PerSAM (K=1, K'=1) | PerSAM (K=5, K'=1) | PerSAM (K=5, K'=5) |
> |----------|----------|----------|----------|
> | 93.3    | 89.3   | 84.5   | 82.9   |
>
> ---
>
> ### W3 and Q2: The application range of the method is limited to patients with multiple exams.
>
> Our work focuses on the patient-specific segmentation task, where prior data can be derived from the initial treatment protocol without imposing additional burden on doctors. Nevertheless, it is worth considering whether random image-mask pairs can be utilized as prior data. If the image-mask pair originates from the same patient, such as another slice in the CT scans, our method can effectively leverage this information. However, if the image-mask pair is unrelated to the current patient, we doubt this strategy would be efficient. In Figures 6, 7, 8, and 9, we showcase qualitative results for follow-up data. In all cases, the follow-up data uses the first-visit or first-frame data as the prior, demonstrating that our method remains robust even when there is an increasing time gap between the follow-up data and the prior data.

---

> > ### Comment · Reviewer_mmQa · 2024-11-25
> >
> > Thank you for the response. They resolved all my questions well.
> >
> > I have a follow-up question on Appendix C. According to the authors' latest response, "the search range is determined by the experimental results". Does this mean the number of positive/negative prompts is determined on the test set? And I'd like to see how robust is the model's performance against this hyperparameter, especially on the Meta/4D-Lung case where there is a significant large amount of negative prompts.

---

> > > ### Author Response · Authors · 2024-11-25
> > > **Response to reviewer mmQa (Part 2)**
> > >
> > > Thank you for agreeing with our arguments in the last response.
> > >
> > > Note that the exact number of positive-point and negative-point prompts for each case is determined by our distribution-similarity-based retrieval approach. Below, we explain how the retrieval (search) range is determined.
> > >
> > > ---
> > >
> > > ### LoRA and Full-Fine-Tune
> > >
> > > - For LoRA and Full-Fine-Tune, the retrieval range is determined based on the validation set of the **i.d.** datasets. We uniformly sample positive-point and negative-point prompts on the ground-truth mask and perform interactive segmentation. The number of prompts is increased until the improvement becomes marginal, at which point this maximum number is set as the retrieval range for **o.o.d.** test datasets.
> > >
> > > - On the 4D-Lung dataset, we consistently set the number of negative-point prompts to 1 for these two types of models. This decision is informed by conclusions from previous works [1], which suggest that the background and semantic target can appear very similar in CT images, and using too many negative-point prompts may confuse the model.
> > >
> > > Table 6 demonstrates that, with our distribution-similarity-based retrieval approach, different retrieval ranges have minimal impact on performance, particularly for in-domain cases (approximately 0.1% on the PerSeg dataset). This aligns with our expectations: the retrieval approach automatically selects the optimal number for the part-aware mechanism based on each case, removing the need to tune it as a hyperparameter.
> > >
> > > [1] Yuhao Huang, Xin Yang, Lian Liu, Han Zhou, Ao Chang, Xinrui Zhou, Rusi Chen, Junxuan Yu,
> > > Jiongquan Chen, Chaoyu Chen, et al. Segment anything model for medical images? Medical
> > > Image Analysis, 92:103061, 2024.
> > >
> > > ---
> > >
> > > ### Meta
> > >
> > > - On the CVC-ClinicDB dataset, the endoscopy video is in RGB space, resulting in a relatively small domain gap compared to SAM's pre-trained dataset. Therefore, for Meta, we use the same retrieval range as the Full-Fine-Tune large model.
> > >
> > > - In contrast, on the 4D-Lung dataset, CT images are in grayscale, leading to a significant domain gap compared to SAM's pre-trained dataset. Consequently, we set the retrieval range for positive-point prompts to 2 to avoid outliers and fixed the number of negative-point prompts to a large constant (i.e., 45) rather than a range, to ensure the model focuses on the semantic target. These values were not further tuned.
> > >
> > > Here, we provide additional results for Meta on the 4D-Lung dataset, showing the impact of setting fewer or more negative-point prompts:
> > >
> > > |   35   |   45  |   55   |
> > > |-------|-------|-------|
> > > |26.49| 28.52 | 30.42 |
> > >
> > > We believe this result is robust, given the significant domain gap between SAM's pretraining dataset and 4D-Lung. Moreover, the performance remains largely unaffected even when 10 negative-point prompts are added or removed.
> > >
> > > ---
> > > We have modified the manuscript in Appendix C (Line 901-917) for clarification based on this question.

---

> > > ### Author Response · Authors · 2024-12-03
> > > **Response to reviewer mmQa (Part 3)**
> > >
> > > We sincerely thank you for your great efforts in reviewing this paper. We have gone through your points one by one and tried to address them carefully. Please don’t hesitate to let us know if you have any further questions.

---

### Official Review · Reviewer_cGra · 2024-11-04

**Soundness:** 2
**Presentation:** 3
**Contribution:** 2
**Rating:** 1
**Confidence:** 5

**Summary:**

The article addresses challenges in personalized treatment within modern precision medicine, particularly in the context of medical image segmentation. The key issues it aims to solve include:

1.	Patient Variability: There is considerable variability among different patients, which complicates the segmentation of tumors and critical organs in medical images.
2.	Limited Annotated Data: Many existing segmentation algorithms rely on large amounts of annotated training data. However, personalized treatment often encounters a shortage of such data for individual patients, making it difficult to train models effectively.

To overcome these obstacles, the authors propose a new approach formulated as an in-context segmentation problem, leveraging the promptable segmentation mechanism of the Segment Anything Model (SAM). Their method, named P2SAM (Part-aware Personalized Segment Anything Model), allows for seamless adaptation to new, out-of-distribution patients using only one-shot patient-specific data.

**Strengths:**

1. The manuscript is clearly expressed and presents the research in a logical and structured manner, although some sentences are lengthy and could benefit from simplification to enhance readability.
2. The paper presents substantial qualitative results and includes experiments across multiple datasets, demonstrating a considerable amount of work.

**Weaknesses:**

1. Representation: The overall logic of the paper appears problematic to me. The title and introduction emphasize "Precision Medicine," yet the writing primarily focuses on highlighting the general applicability of the proposed method. Numerous examples and results are presented from the natural image domain, while the content related to "Precision Medicine" is notably limited, which may cause confusion for readers.

2. The motivation for the "Part-aware Prompt Mechanism" is unclear: what is the reasoning behind this approach, and how does it address challenges in medical tasks? The architecture diagram is also based on natural image applications, leaving it unclear how the proposed method tackles issues specific to precision medicine. Additionally, there is no discussion on how this approach handles different modalities in medical imaging, which the paper should address.

3. Method: Several aspects require further clarification and enhancement.

     a. Firstly, the decision to use only a single negative point per cluster raises concerns regarding its sufficiency. A more robust approach would involve utilizing multiple negative points to enhance model generalization.

     b. Secondly, while the method is designed for patient-specific segmentation, it raises concerns about its application to multi-segmentation tasks. A discussion on how the part-aware prompt mechanism could adapt to scenarios involving multiple segmentations would improve the methodology's applicability.

     c. Additionally, the manuscript relies on 2D segmentation, which requires numerous points for effective performance. This issue remains unaddressed, and the choice not to utilize native 3D models is not sufficiently justified. Given the advantages of 3D models in capturing spatial relationships and providing more comprehensive context in medical imaging, exploring their potential application in this study would strengthen the methodology.

4. Experiments: Several areas require clarification and enhancement.

     a. Firstly, it is unclear whether the number of clusters set for K-means is consistent across different datasets. A detailed explanation of how the clustering parameters are determined for each dataset would improve the robustness of the results and ensure comparability.

     b. Secondly, the comparison methods lack the inclusion of the latest medical-related benchmarks. Integrating more recent studies or state-of-the-art approaches would provide a more comprehensive evaluation of the proposed method's performance. This would help contextualize the results within the current landscape of medical imaging segmentation research.

5. The paper lacks comparisons with state-of-the-art (SOTA) methods. Numerous improvements to SAM tailored for medical imaging have been proposed, yet many of them are omitted in this work’s comparative experiments. Furthermore, several segmentation baselines widely used in the medical domain are not referenced, which casts doubt on the demonstrated effectiveness of the proposed method.

**Questions:**

1. Is the proposed method truly aimed at addressing "Precision Medicine," as suggested in the title and introduction, or is it intended as a more general approach applicable to both natural image and medical imaging domains? These are fundamentally different focuses, and clarity on this point is essential.

2. What is the motivation behind the proposed method, and how does it address some of the fundamental challenges in the medical imaging field, such as unclear lesion boundaries and significant variations across different imaging modalities? Clarity on how these issues are tackled is crucial.

3. Since the method claims to address patient-specific segmentation, it must consider the significant variability in characteristics across different diseases, such as liver cancer, pancreatic cancer, and rectal cancer. How does the proposed method ensure effectiveness when dealing with patients suffering from various diseases with distinct features?

---

> ### Author Response · Authors · 2024-11-18
> **Response to reviewer cGra (Part 1)**
>
> We thank you for your comments and the approval of our target task. In addition to the general response, we address your concerns below:
>
> ### W1 and Q1: The writing logic is problematic.
>
> Our method explicitly aims to address patient-specific segmentation, as highlighted in the title and introduction. We provide the following clarification to rectify your concerns:
>
> - **INTRODUCTION SECTION**: We have emphasized the importance of patient-specific segmentation and its relevance to precision medicine. This establishes the core motivation and focus of our work.
>
> - **METHOD SECTION**:
>   - We have explained the reason we incorporated a natural image in Figure 3, where readers can easily identify each part of the can: characteristics, figures, background, and aluminum material. Feedback from other reviewers indicated that this illustration clarified, rather than confused, the part-aware prompt mechanism. Additionally, we have provided similar illustrations for NSCLC CT scans and polyp images in Figures 12 and 13 in APPENDIX D to show the behavior of the part-aware prompt mechanism on medical images.
>   - The distribution-similarity-based retrieval approach is grounded in the fact that tumors and normal organs exhibit distinct distributions within medical imaging technologies [1] and the observation illustrated in Figure 5.
>   - To ensure the applicability of SAM in the medical imaging domain, we fine-tuned SAM on two medical datasets, as detailed in APPENDIX B.
>
> - **EXPERIMENTS SECTION**: We conducted extensive evaluations on two patient-specific segmentation tasks: NSCLC segmentation for patient-adaptive radiation therapy and polyp segmentation in endoscopy videos. These results are presented in Tables 1, 2, 3, 5, 6, 7, and 8, demonstrating our method’s effectiveness across various patient-specific segmentation applications. While we included results from natural image benchmarks (Table 4) to showcase our method’s versatility across different domains, this aligns with standard experimental protocols to show our method’s generality and does not detract from the patient-specific segmentation problem.
>
> Therefore, we respectfully contend that it is neither accurate nor fair to assert that "the content related to 'Precision Medicine' is notably limited." However, if you have additional suggestions regarding the writing logic, we would be glad to consider incorporating them into the manuscript.
>
> ---
>
> ### W2 and Q2: Motivation behind the proposed method.
>
> The motivation behind the proposed method has been illustrated in the INTRODUCTION SECTION.
>
> - [Lines 050-053]: The part-aware prompt mechanism is introduced to address the ambiguity problem inherent in SAM’s pre-training method. As stated in the SAM paper, “ambiguity is much rarer with multiple prompts,” and our method aims to generate multiple curated prompts to reduce this ambiguity.
>
> - [Lines 078-084]: The distribution-similarity-based retrieval approach is designed to mitigate the effects of occasional outlier prompts generated by the part-aware prompt mechanism. This approach is based on the fact that tumors and normal organs exhibit distinct distributions in medical imaging technologies [1] and is further supported by the observation presented in Figure 5.
>
> We wish to clarify that our method does not primarily aim to address the issue of unclear lesion boundaries, as is the focus of other methods seeking to enhance segmentation performance on internal validation datasets. Instead, our approach is designed to **provide an effective solution for patient-specific segmentation challenges, particularly for out-of-distribution new patients**.
>
> The flexibility of our method naturally results in its adaptability to various imaging modalities. Comprehensive experimental results demonstrate its applicability across diverse patient-specific segmentation tasks, including NSCLC segmentation for patient-adaptive radiation therapy using CT scans and polyp segmentation in endoscopy videos.
>
> ---
>
> [1] Roberto Garc´ıa-Figueiras, Sandra Baleato-Gonz´alez, Anwar R Padhani, Antonio Luna-Alcal´a, Juan Antonio Vallejo-Casas, Evis Sala, Joan C Vilanova, Dow-Mu Koh, Michel Herranz-Carnero, and Herbert Alberto Vargas. How clinical imaging can assess cancer biology. Insights into imaging, 10:1–35, 2019.

---

> ### Author Response · Authors · 2024-11-18
> **Response to reviewer cGra (Part 2)**
>
> ### W3: Method
>
> - Using our distribution-similarity-based retrieval approach, we have explored the research scope for the number of positive-point and negative-point prompts, as shown in Tables 9 and 10 in APPENDIX C. It is important to clarify that employing a single negative-point prompt per cluster results in N negative-point prompts when the background of a medical image is divided into N clusters. Our method is designed this way for the following reasons: 1. While it is possible to use the top-k negative points per cluster, doing so tends to introduce more outliers. 2. Based on findings from previous studies [1], providing an excessive number of negative-point prompts does not enhance model generalization. Instead, it may confuse the model, as the background and the semantic target in medical images can sometimes appear highly similar. To avoid confusion, we have added a discussion about these choices in APPENDIX C. Modifications have been highlighted in blue in the uploaded revised manuscript.
>
> - Our method is evaluated on two datasets: CVC-ClinicDB and 4D-Lung. While the 4D-Lung dataset includes multiple segmentation classes, this work specifically focuses on NSCLC segmentation. Theoretically, our method can be adapted to tasks involving multiple segmentation classes and targets, as illustrated by the example provided in APPENDIX E, Figure 15. Additionally, some datasets listed in Table 4 feature multiple segmentation classes, where our method consistently demonstrates its effectiveness.
>
> - We utilize a 2D model for the following reasons: 1. A 2D model is versatile and can accommodate various medical imaging modalities, such as endoscopy videos and NSCLC CT scans. 2. A 2D model effectively leverages the SAM pre-trained model, significantly enhancing model generalization. This approach also aligns with the strategy employed by MedSAM. Moreover, as discussed in the DISCUSSION SECTION, our method can also integrate with SAM2. This flexibility enables our method to efficiently complement several concurrent works utilizing SAM2 for processing 3D medical image volumes. Finally, when powerful 3D pre-trained backbones become available, our model can be readily adapted to them.
>
> [1] Yuhao Huang, Xin Yang, Lian Liu, Han Zhou, Ao Chang, Xinrui Zhou, Rusi Chen, Junxuan Yu,
> Jiongquan Chen, Chaoyu Chen, et al. Segment anything model for medical images? Medical
> Image Analysis, 92:103061, 2024.
>
> ---
>
> ### W4 and Q3: Lack of experiment settings and comparison with some benchmarks.
>
> - Using our distribution-similarity-based retrieval approach, we explore the research scope for the number of positive-point and negative-point prompts, as shown in Tables 9 and 10 in APPENDIX C. The retrieval range is determined based on experimental results. The search range is determined by the experimental results. Even though, during the experiment, we discovered that the results remained robust even with a wider search range. This can be attributed to our effective distribution-similarity-based retrieval approach. We have included a further discussion about this in APPENDIX C. In Table 6, we also present an ablation study to demonstrate the effectiveness of both the number of clusters and the retrieval approach.
>
> - Segmentation for longitudinal data remains an underexplored area in the community. In this work, we select the two most standard and widely used benchmarks, 4D-Lung and CVC-ClinicDB, which are commonly referenced in related works. Extensive experiments have demonstrated that our method effectively addresses two distinct patient-specific segmentation tasks across various medical imaging modalities: NSCLC segmentation for adaptive radiation therapy using CT scans and polyp segmentation in endoscopy videos. Additionally, we validate the effectiveness of our method on larger and more general natural image datasets. Therefore, we believe that our method can be seamlessly integrated into other imaging modalities and applied to a wide range of diseases. Nevertheless, could you please provide more details about the datasets and benchmarks mentioned in W4 and Q3?
>
> ---

---

> ### Author Response · Authors · 2024-11-18
> **Response to reviewer cGra (Part 3)**
>
> ### W5: Lack of comparison with other methods
>
> - Several studies have adapted SAM for medical image segmentation [1, 2, 3]. However, most of these methods primarily focus on interactively segmenting medical images. In contrast, our work introduces an effective approach that leverages patient-specific prior data to address segmentation for out-of-distribution patients. To the best of our knowledge, none of these studies [1, 2, 3] addresses the specific problem defined in this work.
>
> - We have chosen MedSAM for comparison in Table 3, as it was pre-trained on a large-scale medical image dataset and supplemented with a human-given box prompt during inference. Note that other methods either used smaller pre-training datasets or were not available at the time this work was conducted.
>
> - Consequently, the most relevant baseline methods remain PerSAM and Matcher.
>
> We have added a new section, Appendix F, to discuss these additional related works. Modifications have been highlighted in blue in the uploaded revised manuscript.
>
> [1] Wu, Junde, and Min Xu. "One-prompt to segment all medical images." Proceedings of the IEEE/CVF Conference on Computer Vision and Pattern Recognition. 2024.
>
> [2] Zhu, Jiayuan, et al. "Medical sam 2: Segment medical images as video via segment anything model 2." arXiv preprint arXiv:2408.00874 (2024).
>
> [3] Wong, Hallee E., et al. "Scribbleprompt: Fast and flexible interactive segmentation for any medical image." arXiv preprint arXiv:2312.07381 (2023).
>
> ---

---

> ### Comment · Reviewer_cGra · 2024-11-24
> **Responses to authors**
>
> Thank you for your response. Unfortunately, it still does not fully address my concerns. As you emphasized: the approach is designed to provide an effective solution for patient-specific segmentation challenges, particularly for out-of-distribution new patients. I still have the following questions:
>
> 1. The experimental setup and results do not seem to support the contributions claimed in the paper. Since the goal is to demonstrate that the method can solve patient-specific segmentation problems, why were experiments only conducted in two medical scenarios (such as lung CTs and polyp images), while dedicating significant space to natural image experiments? If the authors claim that the method can segment follow-up data by utilizing prior data as multiple-point prompts, then it should be applicable to other 3D volumes, where a slice could be used to segment follow-up slices. Why wasn't the method tested on more medical segmentation tasks, such as the MSD[1] dataset that includes multiple diseases, to prove its effectiveness? From my perspective, the experimental setup in your paper seems more like an evaluation of a general-purpose method across various domains rather than focusing on solving patient-specific medical image segmentation.
>
> 2. Lack Comparisons with Strong Baselines: I understand that the authors emphasize using one-shot prior data to better segment follow-up patient data. However, there should at least be experimental evidence showing that this approach outperforms some of the current strong baselines in medical imaging, such as nnUNet. These baselines do not consider patient-specific information; they merely use supervised learning to train on limited data and attempt to learn effective features for generalization. The authors need to compare their approach with these baselines and demonstrate that the patient-specific method performs better to justify its significance.
>
> 3. I believe that lesion segmentation is a critical and highly challenging task, characterized by significant variation in morphology, size, and texture, even within the same category of lesions—qualities that differ markedly from those of natural images. Additionally, lesion boundaries are often unclear. These important aspects were not adequately addressed in the experiments presented in the paper, which instead focused on natural image experiments. As a result, I am not convinced that the method can generalize effectively to other diseases. The paper may require further work in this area.
>
> 4. The framework diagram presented in the main paper is still based on natural images. I strongly suggest that the authors consider modifying it to showcase medical images, which would align better with the theme of the paper.  As you mentioned, "We have explained the reason we incorporated a natural image in Figure 3, where readers can easily identify each part of the can: characteristics, figures, background, and aluminum material." Why couldn't a medical image be used for this explanation? Does it mean that the motivation for your method does not originate from issues in medical imaging, making it difficult to illustrate these concepts using medical images? This seems inconsistent with the theme of the paper. Alternatively, as I suggested previously, you could adjust the theme of the paper to propose a segmentation method applicable to multiple domains.
>
> I hope these suggestions can be helpful to the authors.
>
> **Reference**
>
> [1] Antonelli, Michela, et al. "The medical segmentation decathlon." Nature communications 13.1 (2022): 4128.

---

> ### Author Response · Authors · 2024-11-25
> **Response to reviewer cGra (Part 4)**
>
> ### 1. Problem Setting
>
> - We have clearly defined patient-specific segmentation in **Figure 2** and **Section 3.1 (Lines 156–159)**. Specifically, a prior image-mask pair is acquired during the patient’s first visit to the hospital as part of standard clinical protocol. When the patient returns, our method utilizes this existing prior data for segmentation without requiring additional fine-tuning. To evaluate our approach, we selected two medical scenarios: patient-adaptive radiation therapy and polyp segmentation in endoscopy videos. The 4D-Lung dataset, a longitudinal dataset where patients have multiple visits, and the CVC-ClinicDB dataset, an endoscopy video dataset, align well with our problem setting.
>
> - Our method is not specifically designed to address segmentation within a single visit. However, there is no technical obstacle preventing our method from utilizing a labeled slice as prior data to segment an entire 3D image volume. This, however, lies beyond the scope of this paper and could be considered as future work, as discussed in **Section 5 (Lines 531–537)**.
>
> - We do not believe it is detrimental to evaluate our method in the natural image domain to demonstrate its generalization. On the contrary, this evaluation could appeal to a broader audience of readers and researchers, potentially inspiring follow-up ideas in other applications. We respectfully disagree that this is a weakness of our paper; we view it as a strength. Demonstrating the generalization of our method also does not detract from our primary focus on the patient-specific segmentation problem.
>
> ---
>
> ### 2. Strong Baseline
>
> We have compared our method with the current strong baselines in **Table 3**, MedSAM and FCN-Transformers [1]. We choose them as the strong baseline for the following reasons:
>
> - Firstly, we acknowledge that nnUNet is a strong baseline for internal validation (in-distribution). However, we would like to clarify that this paper focuses on the task of external validation (out-of-distribution), where the testing dataset differs from the training dataset. In this scenario, the model’s generalization ability is critical for achieving better performance. Given its pre-training on a large-scale medical image dataset, MedSAM has the potential to outperform many models on external validation sets. While there is no direct evidence to confirm this, DuckNet [2] (Table 1 vs. Table 5 in their paper) demonstrates that large-scale pre-trained models often outperform others on external datasets, even if they lag behind on internal validation sets. This observation is also supported by the authors of nnUNet in nnUnet Revisited [3], who noted that comparing nnUNet with large-scale pre-trained models is unfair.
>
> - The 4D-Lung dataset is a relatively new benchmark for longitudinal data analysis, and no standard benchmark for comparison was available at the time this work was conducted. The results from MedSAM in Table 3 serve as a strong baseline, particularly when supplemented with human-provided box prompts. On CVC-ClinicDB, our direct-transfer baseline outperforms FCN-Transformers, which achieved the best result reported in DuckNet [2] under the same evaluation objective: training on Kvasir-SEG and testing on CVC-ClinicDB.
>
> [1] Sanderson, E., & Matuszewski, B. J. FCN-transformer feature fusion for polyp segmentation. In Medical Image Understanding and Analysis 892–907. Springer. https://doi.org/10.1007/978-3-031-12053-4_65 (2022).
>
> [2] Dumitru, Razvan-Gabriel, Darius Peteleaza, and Catalin Craciun. "Using DUCK-Net for polyp image segmentation." Scientific reports 13.1 (2023): 9803.
>
> [3]Isensee, Fabian, et al. "nnu-net revisited: A call for rigorous validation in 3d medical image segmentation." International Conference on Medical Image Computing and Computer-Assisted Intervention. Cham: Springer Nature Switzerland, 2024.

---

> ### Author Response · Authors · 2024-11-25
> **Response to reviewer cGra (Part 5)**
>
> ### 3. Unclear Boundary
>
> - Non-small cell lung cancer (NSCLC) segmentation is already a challenging task. The last two columns in **Figure 8 (Section 4.3)** illustrate an example where the tumor boundary is unclear. The direct-transfer baseline struggles in this scenario, whereas our method effectively alleviates this issue by efficiently extracting relevant information from the prior data to prompt the current segmentation.
>
> - We do not believe that demonstrating our method’s generalization ability on standard natural image benchmarks detracts from our contribution to the proposed patient-specific segmentation problem. In **Section 4.2**, we perform experiments incorporating **various transfer learning methods (Meta, LoRA, full fine-tune)**, **model sizes (base and large)**, and **approaches (direct-transfer, fine-tune, PANet, Matcher, PerSAM)** on two medical scenarios, as shown in **Tables 1 and 2**. We also compare our method with current strong baselines in **Table 3**. **In Section 4.4**, we conduct a comprehensive ablation study **(Tables 6, 7, and 8)** on the CVC-ClinicDB dataset to evaluate the effectiveness of each module in our method. These experiments align closely with our focus on the patient-specific segmentation problem. The comparison of natural image benchmarks (Table 4) is limited to standard evaluations. As a result, over **90%** of our experiments are conducted on medical segmentation tasks.
>
> ---
>
> ### 4. Framework Figure
>
> - It is possible to present the framework figure using medical images as an example. We have already included medical image examples in **Figure 12 and Figure 13 (Appendix D)** and provided interpretations for each part in **Appendix D (Lines 963–965)**. For both NSCLC and polyp segmentation, there is no explicit definition when dividing the tumor into several parts; these parts are more "data-driven." However, they still provide valuable insights into why the approach improves performance. As discussed in **Section 4.3 (Lines 421–423)**, these parts contribute by offering precise foreground information when the target is too small and additional context when the segmentation is incomplete.
>
> - Considering the need to communicate with a general audience that may lack a background in medical imaging, we believe that using a natural image example is more straightforward for illustrating the key points of our proposed method. This choice is purely for illustration purposes and should not be confused with the motivation behind our work.

---

> > ### Comment · Reviewer_cGra · 2024-11-28
> > **Further Response to Authors**
> >
> > I do not think the authors have adequately addressed my concerns. Please note that I am not suggesting that validating the method on natural images is detrimental. My concern is that the validation in the medical domain and the experimental setup are insufficient to support the claimed contributions. I do not believe this paper can be said to have "addressed an important task—patient-specific segmentation," given that many key scenarios across different anatomical regions were not evaluated. Therefore, compared to the missing experiments, validation on natural images is far less important than conducting validation across more medical scenarios.
> >
> > I hope the authors can provide more substantial experimental validation in the medical imaging domain, but instead, they continue to argue for the effectiveness of their method in the natural image domain. Ultimately, this paper appears more like a general-purpose model designed for multiple domains rather than specifically for patient-specific segmentation in medical imaging. If the authors continue to assert that the paper's focus and contribution lie in patient-specific segmentation within the medical field, then I believe this work clearly requires further effort and is not ready for publication at this stage. I also do not think that comparisons with methods like MedSAM or FCN-Transformers are sufficient to demonstrate the superiority of the approach.
> >
> > Furthermore, the paper emphasizes the out-of-distribution generalization capability of the proposed method but lacks comparisons with state-of-the-art (SOTA) methods in this regard. For these reasons, I choose to maintain my score.

---

> ### Author Response · Authors · 2024-12-04
> **Response to reviewer cGra (Part 6)**
>
> ---
>
> Regarding your concerns:
>
> - Motivation behind the proposed method.
>
> - Each cluster corresponds to only one negative-point prompt.
>
> - Experiment settings for the number of clusters.
>
> - Comparison with existing strong baselines.
>
> We have addressed these points carefully. We hope these answers have resolved these issues for you.
>
> ---
>
> We regret that we were unable to address some of your other concerns, as we believe they extend beyond the scope of this article. Regarding additional medical scenarios, the MSD dataset does not contain longitudinal data and therefore does not align with the problem this article aims to address. Moreover, while nnUNet is a robust method for most supervised medical image segmentation tasks, the authors have stated in [1] that it is not fair to compare nnUNet with recent large-scale pre-trained methods. Therefore, we believe nnUNet is not designed for the out-of-distribution evaluation.
>
> We treat MedSAM and FCN-Transformer as strong baselines because MedSAM was pre-trained on a large-scale medical dataset. In a state-of-the-art method for supervised polyp segmentation, DuckNet [2], FCN-Transformer [3] was reported as the state-of-the-art method under the same evaluation objective.
>
> [1]Isensee, Fabian, et al. "nnu-net revisited: A call for rigorous validation in 3d medical image segmentation." International Conference on Medical Image Computing and Computer-Assisted Intervention. Cham: Springer Nature Switzerland, 2024.
>
> [2] Sanderson, E., & Matuszewski, B. J. FCN-transformer feature fusion for polyp segmentation. In Medical Image Understanding and Analysis 892–907. Springer. https://doi.org/10.1007/978-3-031-12053-4_65 (2022).
>
> [3] Dumitru, Razvan-Gabriel, Darius Peteleaza, and Catalin Craciun. "Using DUCK-Net for polyp image segmentation." Scientific reports 13.1 (2023): 9803.
>
> ---
>
> We sincerely thank you again for your great efforts in reviewing this paper.
>
> ---

---

### Author Response · Authors · 2024-11-28
**General Response and Summary of Updates to Manuscript**

We thank the reviewers for noting that we addressed an important task—patient-specific segmentation—with a straightforward and effective method (mmQa, 3qfj, 9LGG) and further achieved impressive generalization performance across various segmentation tasks and domains (mmQa, 3qfj, 9LGG). First, we provide a summary of the changes that we have made to the manuscript to address your feedback and conclude with an overview of our key contributions.

---

Here is a summary of the updates that we have made to the manuscript:

- Added more implementation details of the strategy and reason we chose the retrieval range in Appendix C. (mmQa, cGra)

- Added an appendix section (Appendix F) to discuss additional related works (UniverSeg, Med-SA, One-Prompt, ScribblePrompt) and the reason we chose MedSAM as the strong baseline in this work. (9LGG, cGra)

- Added the abovementioned works to Section 2 and Section 3. (9LGG)

- Clarified we treat MedSAM and FCN-Transformers as a strong baseline instead of the state-of-the-art method in Section 4.2 and Table 3. (9LGG, mmQa)

---

To contextualize the results in this paper, we now summarize our contributions and insights:

- **Solving a challenging problem.** Effectively utilizing the limited prior information in longitudinal medical data analysis for each specific patient is a challenging task. Instead of relying on traditional transfer learning methods that fine-tune models on limited prior data for individual patients, we propose an in-context segmentation solution tailored for patient-specific segmentation.

- **Investigating the promptable mechanism.** The promptable mechanism is a key module for foundation models built upon the human-in-the-loop strategy. However, a simple strategy for applying this mechanism to downstream tasks can cause ambiguity and outliers prompts, and result in bad performance. In this work, we propose a part-aware prompt mechanism to generate diverse and robust prompts.

- **Proposing a distribution-similarity-based retrieval approach.** Given that tumors and normal organs exhibit distinct distributions within medical imaging technologies (see Figure 5), we propose a distribution-similarity-based retrieval approach to determine the optimal number of parts required for individual cases. This retrieval approach is non-trivial and crucial.

- **Impressive performance.** Our work has achieved impressive performance across multiple segmentation tasks, and extensive ablation studies have confirmed that the proposed components are effective and necessary.

---

### Meta-Review · Area_Chair_ggUQ · 2024-12-20

**Metareview:**

Key issues:
1. Mismatch between focus and title.
2. Methodology clarification needed.
3. Incomplete comparisons.
4. Technical concerns:
- The clustering parameters' determination across different datasets needs explanation
- The computational demands of your approach should be addressed
- The method's applicability to diverse medical imaging modalities requires discussion

**Additional Comments On Reviewer Discussion:**

The reviewers acknowledge the paper's clear writing and thorough experimentation. However, addressing these concerns, particularly the comparison with recent work and strengthening the medical imaging focus, is crucial for advancing the manuscript.

---

### Decision · Program_Chairs · 2025-01-22

Reject